# Self-spinning filaments for autonomously linked microfibers

Dylan M. Barber[1], Todd Emrick [1]✉, Gregory M. Grason [1]✉ & Alfred J. Crosby [1]✉

Filamentous bundles are ubiquitous in Nature, achieving highly adaptive functions and structural integrity from assembly of diverse mesoscale supramolecular elements. Engineering routes to synthetic, topologically integrated analogs demands precisely coordinated control of multiple filaments' shapes and positions, a major challenge when performed without complex machinery or labor-intensive processing. Here, we demonstrate a photocreasing design that encodes local curvature and twist into mesoscale polymer filaments, enabling their programmed transformation into target 3-dimensional geometries. Importantly, patterned photocreasing of filament arrays drives autonomous spinning to form linked filament bundles that are highly entangled and structurally robust. In individual filaments, photocreases unlock paths to arbitrary, 3-dimensional curves in space. Collectively, photocrease-mediated bundling establishes a transformative paradigm enabling smart, self-assembled mesostructures that mimic performance-differentiating structures in Nature (e.g., tendon and muscle fiber) and the macro-engineered world (e.g., rope).

The power of self-organization in Nature gives rise to sophisticated, hierarchical structures with remarkable properties[1,2]. Indeed, Nature's mesoscale building blocks, spanning 100 nm–10 μm, are essential for bridging length scales, from nano/molecular to the macroscale. Crucially, the assembly of mesoscale building blocks presents a unique challenge because individual units are too large to sample the energy landscape via thermal motion, yet small enough that macroscale assembly methods are impractical[3,4]. In particular, aligned arrays of mesoscale fibrils—a class of structures underpinning a vast selection of biomaterials—possess few synthetic analogs. Key natural examples include collagen in load-bearing tendon and bone[5–10]. actuating muscle[11–17], and plant structures that simultaneously provide strength, flexibility, and nutrient transport[18–22]. In these systems, unique, anisotropic properties are derived from organization and collective behavior of aligned mesoscale filaments.

Our design strategy maps a route to assemblies of synthetic mesoscale filaments by considering key examples of fiber assembly that hinge on self-organization at different length scales. In one case,

proteins like fibrin, responsible for blood clotting at wound sites[23,24], and sickle-cell hemoglobin[25], self-organize into coaxial, aligned multifilament fibrils. Intermolecular chiral interactions between biomolecular subunits may promote a tilt between successive layers to drive a global twist in the bundles. In this class of supramolecular bundles, assemblies are maintained by cohesive interactions between distinct building-blocks, enabling (for example) rapid gelation to mitigate blood loss. Another example of filamentous assembly is found at the macroscale, arising from top-down spinning of fibrous materials like cotton and wool. In these structures, individual strands are collectively twisted in one direction, which propagates strand linking into a structurally coherent yarn, which in turn may be aligned and twisted together to form a hierarchical rope[26,27]. Due to this twisted bundle architecture, a tensile load creates lateral compression to dramatically increase inter-filament friction and prevent slip[28–30]. The remarkable 'self-reinforcing' outcome of this process serves to transfer forces over lengths that dramatically exceed those of the constituent filaments. In contrast to the cohesive secondary interactions that mediate fibrin

[1]Polymer Science and Engineering Department, University of Massachusetts Amherst, Amherst, MA 01003-9263, USA.
✉e-mail: tsemrick@mail.pse.umass.edu; grason@mail.pse.umass.edu; acrosby@umass.edu

assemblies, the key feature maintaining a yarn is its kinematically derived topological linking, in which entanglement, rather than interactions, underpins their properties.

At the mesoscale, both assembly modes (i.e., fibrin-like, interaction-mediated; and yarn-like, topology-mediated) offer exciting pathways to design types of bundled structures. An ideal example of an interaction-mediated system would use site-selective fibrillar binding to facilitate ordered growth. However, while recent reports describe site-specific compositions in mesoscale building blocks[31–39] (including mesoscale analogs of block copolymers[40,41]), regioselective interactions in synthetic filaments lack the sophistication to realize spontaneous meso-to-macroscale assembly and long-range order. For example, capillary attraction[42,43] was shown by Pokroy, et al. to drive micropillar assembly in the absence of regiospecific interactions[44]. In their system, template-derived[45] micropillars were drawn into bundles via capillary forces and held in place by interpillar cohesion to afford twisted assemblies. Yet, this system lacks site-specificity, and it is unclear whether the assemblies have sufficient cohesion to realize a robust and free-standing structure. Moreover, the maximum length of individual pillars was 9 μm, suggesting mechanically negligible entanglement or links between bundled pillars. With regards to the yarn-like (link-mediated) case, reports of mesofilament assembly are limited to disordered arrays. For example, the acrylate gel fibers of Perazzo, et al. exhibit shear-thickening behavior that the authors attribute to interfibrillar entanglement[46]. In this case, the assembled gel lacks the order and anisotropy of biomaterials like bone and tendon. Thus, the twisted, topologically interlinked mesoscale bundle remains an unrealized target. To this end, we introduce a route to self-assembled mesoscale filaments underpinned by generalizable principles. Our materials platform consists of mesoscale polymer (MSP) filaments with programmed local (i.e., arc-length-dependent) curvature and twist such that, upon application of a stimulus, each filament deforms to trace a predetermined 3D path in space, an accomplishment that has been confined to the macroscale[47,48] before this work. Importantly, this programming of curvature and twist is achieved by locally applying the well-understood principles of deformation mismatch through the thickness of a beam (in this case, controlled in-

plane differential swelling, Supplementary Fig. 1 and Ref.[49]). Thus, while the selected chemistry delivers the target assemblies, the fabrication pathway described below is, in principle, readily generalizable across any material compositions that enable deformation mismatch upon application of a stimulus. As described below, by tailoring the timing and location of assembly conditions, multiple filaments self-bundle in a programmed manner to realize structurally robust assemblies that are held together by a high density of inter-filament linking.

## Results

Our approach begins with arrays of discrete MSPs prepared by a solution-phase fabrication method termed flow-coating[50–52]. The MSPs are embedded with photo-induced creases that direct local fold-like deformation between stiff segments along the MSP length; we utilized oriented photocreases to encode constant twist and curvature, affording a segmented helix of defined pitch, radius, and handedness. Under appropriate conditions of inter-MSP spacing and photocrease deformation dynamics, we found that photocreased MSP arrays collectively spin into roughly cylindrical twisted bundles with hollow centers (inner radius ~18–20 μm) and a variable outer radius (spanning ~24–55 μm) that depends on the number $n$ of ribbons per bundle and the photocrease 'tilt angle' $\phi$. Importantly, these twisted MSP bundles display enormous topological entanglement, with a linking number >200 for a 10-MSP bundle of ~500 μm length[53]. Moreover, MSPs confined within bundles exhibit larger helical pitch $p$ and radius $r$ relative to single (unbundled) MSPs, suggesting a collective compressive stress that mimics the lateral gripping effect of a yarn under tension. Such structural features may enhance the integrity of the resultant material, as described later when interlinked bundles are visualized in a liquid with an applied flow.

### Filaments for programmed kinematic control
Self-assembly of a twisted bundle requires embedding the constituent filaments with the ability to undergo programmed kinematic transformations. Our design strategy, outlined in Fig. 1, exploits parallel alignment of MSP ribbons (length $L$, width $w$, thickness $h$) with period

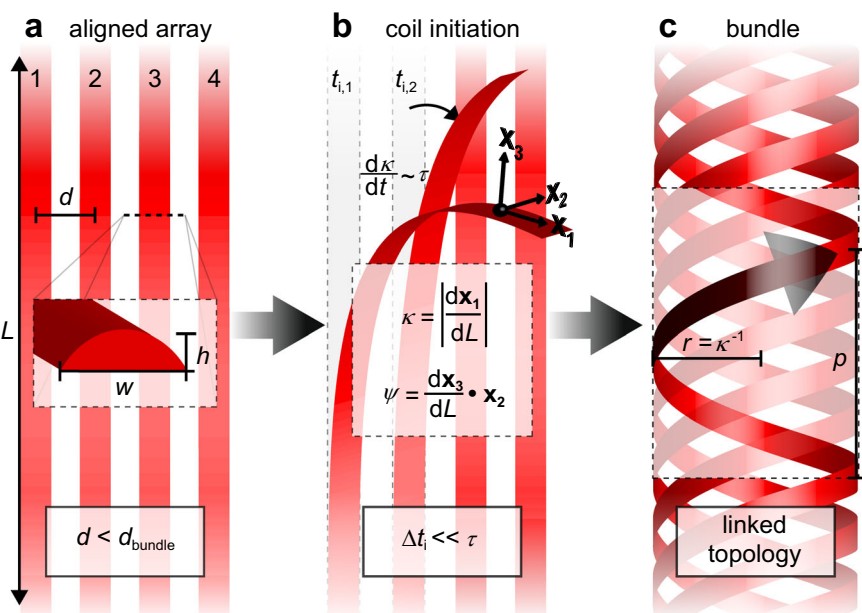

**Fig. 1 | Key parameters in twisted bundle formation. a** Aligned ribbons of thickness $h$, width $w$, and arc length $L$ with periodic spacing $d$; **b** coil initiation of ribbons 1 and 2 at times $t_{i,n}$. The timescale $\tau$ defines coiling rate after initiation with

curvature $\kappa$, twist $\psi$, and arc-length-dependent orthonormal frame $\{\mathbf{x_1},\mathbf{x_2},\mathbf{x_3}\}$; **c** bundled ribbons with constant $\kappa$ and $\psi$ possess uniform chirality, radius $r$, and pitch $p$.

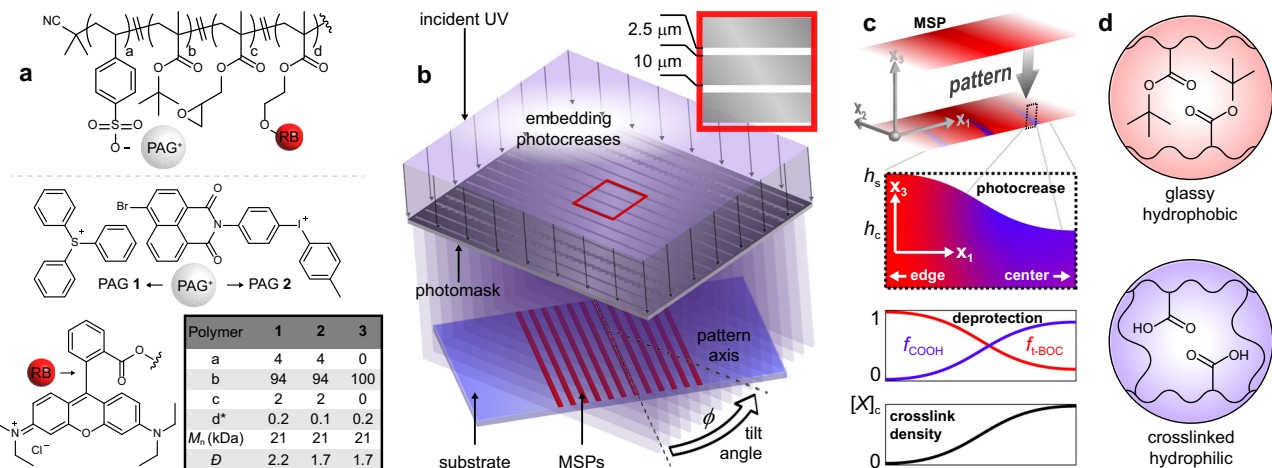

**Fig. 2 | Strategy for embedding photocreases into MSPs. a** Random copolymer structure used in this study, where the counter ion is sulfonium (PAG **1**) or iodonium (PAG **2**) and RB represents the rhodamine B fluorophore. Incorporation ratios **a**–**c** were determined by ¹H NMR peak integration while **d** was estimated from the feed ratio; molecular weight distributions were estimated by gel permeation chromatography; **b** schematic depiction of MSP patterning to embed photocreases with tilt angle $\phi$; **c** evolution of photocreases during patterning. Inset schematics: MSP thickness decreases along the $x_1$ direction from the edge to the center of the photocrease; **d** color coding of local hydrophobic (unreacted) and hydrophilic (crosslinked) MSP components.

separation distance $d$ (Fig. 1a). The MSPs are then structurally altered to promote coiling into helices of uniform curvature ($\kappa$) and twist ($\psi$), reflected as constant chirality, radius ($r = \kappa^{-1}$), and pitch $p$. Both $\kappa$ and $\psi$ are defined in terms of the arc length-parametrized frame {$x_1, x_2, x_3$}, corresponding to the tangent, normal, and binormal vectors to the MSP centerline, respectively. MSP release from the substrate initiates coiling at time $t_{i,n}$. Upon MSP release, the rate of coiling, given by the change of $\kappa$ with time, is governed by the timescale $\tau$, while MSP spacing and release kinetics dictate bundle formation. Specifically, bundling occurs when the inter-MSP spacing is sufficiently small and the initiation time interval is much shorter than the coiling timescale ($\Delta t_i = t_{i,n+1} - t_{i,n} \ll \tau$). Moreover, the constituent MSPs must adopt homochiral shapes (Fig. 1c), not unlike analogous biofilaments.

To program MSP structures, we prepared random copolymers with embedded functionality sensitive to light (Fig. 2a) that give access to MSPs with defined regions of decreased thickness and increased hydrophilicity that are adjacent to thicker, more rigid hydrophobic regions. Copolymers **1** and **2** (Fig. 2a) were prepared with *t*-butyl methacrylate (TBMA) as the major component, with small amounts of glycidyl methacrylate (GMA) to promote cross-linking and triphenylsulfonium (PAG **1**) or naphthalimide-substituted diaryliodonium (PAG **2**) as photoacid generators, in which the PAGs were integrated into the polymer structures as counter ions rather than as separate, small molecule additives[54–59]. Copolymers **1** and **2** were prepared by conventional free radical polymerization to afford random copolymers and afford uniform distribution of each pendent group through the printed MSP bulk. In addition, there was no preference or improved performance between copolymers **1** and **2**; the photocrease platform was demonstrated with both PAGs to emphasize its versatility and generalizability. Polymer **3**, a TBMA homopolymer, was employed in control experiments. Preparing these polymers with trace (0.1–0.2 mole percent) rhodamine B-substituted methacrylate enabled easy visualization by fluorescence microscopy. Further rationale for mole ratios of selected monomers are discussed in Supplementary Note 3.

These photoactive polymers were deposited by flow-coating from a toluene solution to afford aligned MSP arrays on a poly(styrene sulfonate) (PSS)-coated glass substrate, then were irradiated through a photomask (Fig. 2b) and heated to 150 °C to produce alternating domain widths of 2.5 μm (irradiated, hydrophilic) and 10 μm (masked, hydrophobic). The irradiated domains are referred to as 'photocreases,' and the tilt angle $\phi$ denotes the alignment between the substrate-bound MSPs and the photomask line-space pattern; $\phi = 0°$

and $\phi = 90°$ describe photocreases situated parallel and perpendicular, respectively, to the MSP long axis ($x_1$). Fig 2c describes the geometrical and compositional transformations associated with patterning. A uniform MSP (top) is irradiated through a photomask to afford photocreases. The inset depicts the thickness change of the photocrease along the $x_1$ direction from a maximum of $h_s$, the glassy segment thickness at the photocrease edge, to $h_c$, the thickness at the photocrease center. The decrease in thickness arises from deprotection of pendent *t*-butyl esters and mass loss from the polymer. Accordingly, the thickness gradient correlates with the extent of deprotection, described by $f_{COOH} = 1 - f_{t-BOC} = \frac{[COOH]}{[COOH]+[t-BOC]}$ (where [z] is the concentration of functional group z in the MSP structure). Simultaneously, the pendent epoxide groups undergo acid-catalyzed crosslinking to afford local polymer networks (Fig. 2c, bottom) with an anticipated gradient in crosslink density starting from 0 at the mask edge to a maximum of $[X]_c$ at the center of the irradiated region. Thus, the gradient thickness, measured by optical profilometry, indicates a smooth transition from thick, glassy, and hydrophobic PTBMA at the photocrease edge to thin, crosslinked, and hydrophilic poly(methacrylic acid) (PMAA) at the photocrease center (color-coded schematics Fig. 2d). Additionally, attenuation of incident UV radiation through the photocrease thickness affords a gradient in activated photoacid in the $x_3$ direction, resulting in a through-thickness gradient of $f_{COOH}$ that in turn affords a swelling gradient upon introduction of aqueous solution, bending the photocreases locally (Supplementary Fig. 1). The use of differential swelling through the thickness of a beam (in this case, an MSP) is well understood in the design of 3D shape-morphing structures[49]. As we detail below, the combination of the local folding along the filament at photocreases in combination with the photolithographic control over the angle and spacing of those patterned folds, affords the ability to effectively program the 3D shape ($\kappa$ and $\psi$) of the ultimate filaments. Accordingly, the strategy of encoding stimulus-responsive creases along the filament length is readily generalizable across a range of chemistries and is not limited to the chemically amplified resist compositions employed in this work.

**Programming 3D MSP geometry**

Figure 3a depicts substrate-bound MSPs with embedded photocreases at angles $\phi$ and the structures anticipated upon release from the substrate, with $\phi$- and mask-dependent photocrease length $l_c$ and segment length $l_s$. Assuming that the photocreases in a given MSP bend, or fold, in the same direction (i.e., bottom-face-in or bottom-

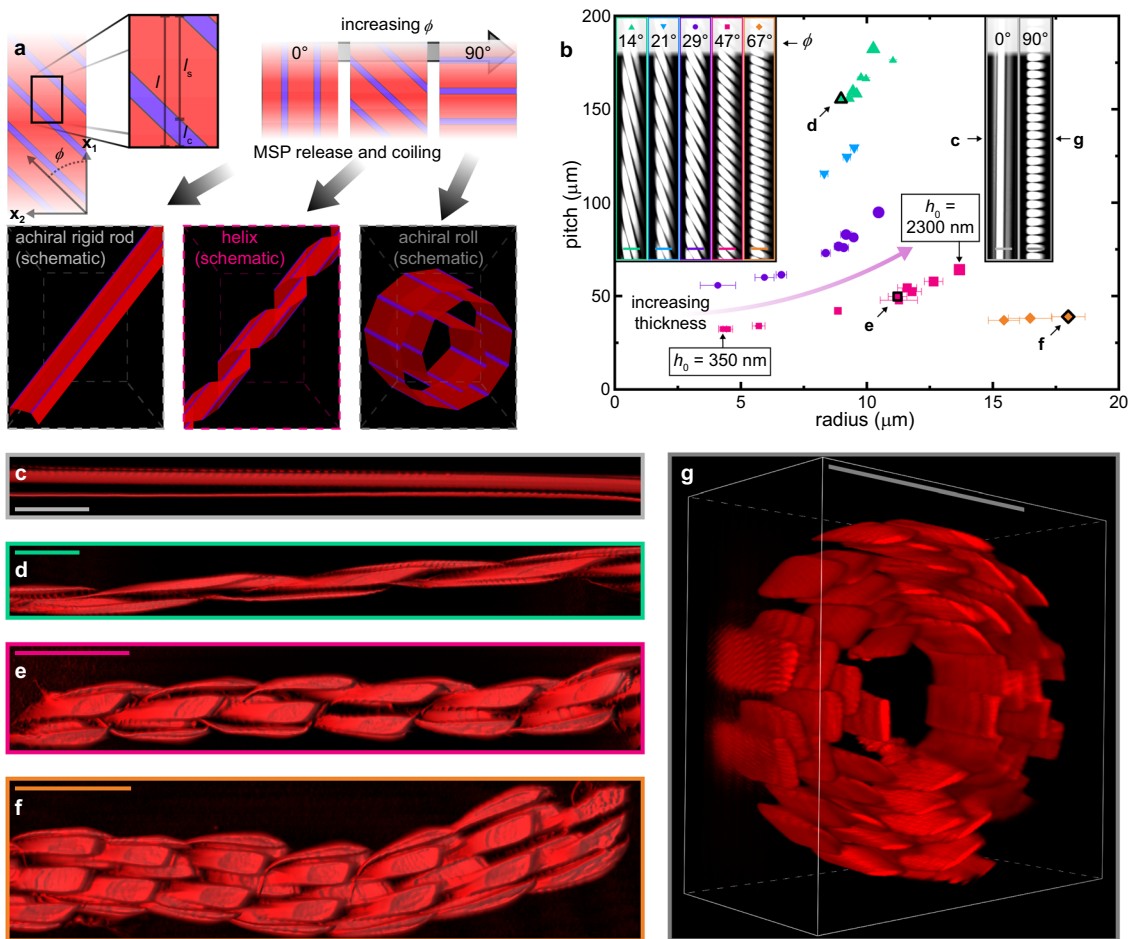

**Fig. 3 | MSP coiling. a** schematic of anticipated 3D structures arising from photocrease bending for $\phi = 0°$ (left), $0° < \phi < 90°$ (center), and $\phi = 90°$ (right); only intermediate values of $\phi$ are expected to afford helices. **b** helix pitch as a function of helix radius for $\phi = 14°$, $21°$, $29°$, $47°$, and $67°$. For $\phi = 29°$ and $47°$, increasing thickness ($h_0$ -350–2300 nm) affords increasing pitch and radius; data point size scales with $h_0$. Error bars represent 1 standard deviation of the sample summed in quadrature with rms displacement from fitted helix; 3 highlighted data points correspond to the helices in **d–f**. Inset: fluorescence micrographs of substrate-bound MSPs before release, including 5 prochiral (left) and 2 achiral (right, corresponding to **c** and **g**) MSPs; inset scale bars 25 μm. **c**, a released MSP with $\phi = 0°$ affords an achiral rigid rod, while $\phi = 14°$ (**d**), $47°$ (**e**), and $67°$ (**f**) afford helices and $\phi = 90°$ (**g**) affords an achiral roll, with structure edge cropped to reveal cross-sectional profile. Scale bars in **c–g** are 50 μm; color coding in **c-g** is consistent with **b**.

face-out), the resultant 3D geometries depend on $\phi$ in three key regimes. When $\phi = 0°$, the photocreases are parallel to the MSP and bending affords a rigid rod with the same length as the undeformed MSP in the substrate-bound condition. At the opposite extreme, $\phi = 90°$ is anticipated to produce 'rolled' structures by photocrease bending such that a fraction of photocreases adopt a preferred fold and the remaining photocreases are sterically constrained to shallower fold angles. Intermediate values ($0° < \phi < 90°$) break the mirror symmetry of the MSP, affording prochiral structures that set up helix formation upon release. In this case, $\phi$ will determine the effective twist and curvature of the helices, resulting in controlled pitch, radius, and handedness; $\phi \to 0°$ is anticipated to induce large pitch and small radius, while $\phi \to 90°$ will effect a small pitch and large radius.

To test this dimensional control, photocreased MSP arrays were fabricated by flow-coating copolymer **1** or **2**, followed by photopatterning, then released from the substrate by dissolving the underlying sacrificial layer in water. MSPs were characterized before and after release by conventional and confocal fluorescence microscopy methods. Seven samples of copolymer **2** MSPs of varying thickness are represented by data points and 3D-reconstructed confocal z-stacks in Fig. 3b–g. These were patterned at a UV dose of 75 J cm$^{-2}$ and $\lambda_{max} = 365$ nm, then heated to 150 °C for 60 s to embed photocreases with $\phi = 0°$, $14°$, $21°$, $29°$, $47°$, $67°$, and $90°$. The MSPs were released into a pH 8

buffer solution to initiate photocrease bending. Upon release, the MSPs in the five samples with intermediate values (i.e., $\phi = 14°$, $21°$, $29°$, $47°$, and $67°$) coiled within 10 s into stable and uniform left-handed helices. The pitch and radius of each helix was determined by least-squares fitting to the centroids for each glassy PTBMA segment[60,61], and were consistent with the qualitative predictions outlined above. Specifically, smaller values of $\phi$ afforded stiffer helices that showed less curvature along -mm-scale length, as well as larger pitch and smaller radius. Moreover, samples with $\phi = 14°$, $29°$, and $47°$ were used to investigate the role of thickness in determining $p$ and $r$, with as-printed MSP thickness $h_0$ spanning 460–2050 nm, 440–2030 nm, and 350–2300 nm across 8, 9, and 10 helices, respectively. Interestingly, pitch and radius correlated positively with $h_0$ for the $\phi = 29°$ and $47°$ cases, but appeared independent of thickness for $\phi = 14°$, suggesting some interdependence between $\phi$ and thickness in determining helix dimensions. Supplementary Figs. 2 and 3, respectively, plot helix radius and pitch as a function of $h_0$ for each of the five chiral samples in Fig. 3b. Supplementary Fig. 4 collapses the data of Fig. 3b by plotting the same data set as $p(\text{Tan}(\phi))$ as a function of radius. The insets in Fig. 3b are grayscale fluorescence micrographs of representative substrate-bound MSPs of $h_0$ -1 μm after photopatterning. Conveniently, the pendent rhodamine B fluorophore partially quenched during photopatterning by photoacid-mediated quenching, enabling

facile distinction between exposed and unexposed regions of the MSPs.

Figure 3c–g show 3D-reconstructed confocal z-stacks of representative MSPs with $h_0$ -1 μm after release and photocrease bending. Notably, MSPs with $\phi = 0°$ (Fig. 3c) afforded a straight, rigid rod (as predicted in Fig. 3a), a geometry not conducive to bundle formation. Similarly, Fig. 3g shows an achiral roll (cropped in plane to highlight individual segments; Supplementary Fig. 5 shows an uncropped structure). In contrast, MSPs with $\phi = 14°$ (Fig. 3d), 21°, 29°, 47° (Fig. 3e), and 67° (Fig. 3f) proved amenable to helix formation upon release into the fluid phase. For $\phi = 14°$, narrow helices formed ($r$ -9–11 μm) with large pitch (155–176 μm) and negligible curvature along their -560 μm length. However, increasing $\phi$ resulted in a smaller pitch, until the adjacent helical coils overlapped, resulting in steric hindrance and decreased helix uniformity. The resulting helices showed shorter pitch and larger radius, deviating from a uniform cylinder on much shorter length scales than the $\phi = 14°$ case. This was particularly pronounced for $\phi = 67°$ (Fig. 3f; note larger uncertainty in radius compared to other values of $\phi$), in which the helix shows significant deflection of its central axis along the -250 μm length captured in the confocal z-stack of Fig. 3f. Interestingly, the helices responded to fluid flow introduced by injecting liquid with a pipette in a similar manner, with $\phi = 14°$ helices behaving as rigid rods and $\phi = 67°$ showing greater flexibility. Finally, control of curvature and twist implies programmable chirality, as found by changing the sign of $\phi$ (i.e., $\phi < 0°$ vs. $\phi > 0°$) to encode helix handedness, as described in Supplementary Note 1, Supplementary Fig. 6, and Supplementary Movies 1 and 2.

## Mechanistic insights

A series of experiments was conducted to understand the variables driving photocrease bending and helix formation (Supplementary Figs. 7–9), as well as support the proposed differential swelling mechanism upon introduction of aqueous solution (Supplementary Fig. 1, Supplementary Fig. 6). First, given that photocreases are composed of crosslinked PMAA, the role of photocrease swelling was characterized by releasing MSPs into buffers of varying pH (Supplementary Fig. 7). For example, an MSP array was released into an aqueous buffer at pH 1, which minimized photocrease swelling by protonating carboxylate groups and decreasing the hydrophilicity of the photocreases; a representative MSP with $h_0 = 1.05$ μm and $\phi = 41°$ showed a pitch of 241 μm and a radius of 32 μm (image Supplementary Fig. 7a, plotted Supplementary Fig. 9), a -3-5-fold increase over the dimensions observed in pH 8 buffer with identical MSP thickness and 47° tilt angle ($p = 50$ μm, $r = 11$ μm, Fig. 3b, e). Upon changing the buffered medium from pH 1 to 10, which deprotonates and swells the PMAA network in the photocreased regions, bending progressed to higher fold angles to afford helix dimensions consistent with those observed at pH 8 (image Supplementary Fig. 7c, plotted Supplementary Fig. 9) and suggesting that swelling plays a significant role in photocrease bending. In another experiment, an MSP array ($\phi = 44°$) was released into pH 10 buffer, imaged, then transferred to a pH 1 buffer. A representative MSP with $h_0 = 1.08$ μm immediately coiled to 11 μm radius and 51 μm pitch at pH 10 (image Supplementary Fig. 7c, plotted Supplementary Fig. 9); this did not change significantly upon acidification to de-swell the photocreases (image Supplementary Fig. 7d, plotted Supplementary Fig. 9), suggesting irreversibility in photocrease bending that may be attributed to the existence of a lightly crosslinked region in the photocrease. Next, the role of interfacial tension at pH 8 was tested when an MSP array ($\phi = 47°$) was released into pH 8 buffer containing 3 mM sodium dodecyl sulfate (SDS, image Supplementary Fig. 8, plotted Supplementary Fig. 9). A representative MSP with $h_0 = 1.04$ μm coiled to a 14 μm radius and 60 μm pitch, slightly larger than the observed surfactant-free dimensions, suggesting that interfacial tension has a slight impact on MSP coiling for $h_0$ -1 μm. Significantly, interfacial energy has been shown to

drive bending, due to the cross-sectional asymmetry inherent to flow-coated ribbons[51,62,63]. This phenomenon was decoupled from photocrease folding by preparing a copolymer 2 film of $h_0$ -1 μm, then irradiating (75 J cm⁻² and $\lambda_{max} = 365$ nm) and heating (60 s, 150 °C) to fully deprotect it. The film was laser cut at 500 μm intervals to afford ribbon-like strips of deprotected PMAA film with a uniform cross-section. Upon release into pH 8 buffer, the structures rolled into long tubes over the first -30 s after release, then elongated to -1.85x their dry length. Confocal microscopy confirmed the bending direction in these cut-film helices to be bottom-face-in, consistent with the bending direction observed in photocreases (Supplementary Fig. 10). Together, these experiments reveal that swelling is critical to the photocrease bending mechanism, localized folding at photocreases is the primary mechanism of helix formation (with a negligible contribution from surface tension), bottom-face-in curvature persists in fully deprotected thin films, suggesting that this curvature does not depend on cross-sectional asymmetry or material gradients in the $\mathbf{x_1}$ (tangent) direction of MSPs, and photocrease bending is driven by a through-thickness swelling gradient, with greater swelling near the top surface.

The above experiments confirm that curvature in photocreases and deprotected polymer films is swelling-mediated and occurs in a bottom-face-in orientation. The attenuation of the incident UV radiation through the MSP thickness produces a photoacid gradient in the $\mathbf{x_3}$ direction. This creates a gradient of deprotected carboxylic acid (i.e., relative hydrophilicity and amenability to swelling) through the photocrease thickness. Upon release into an aqueous buffer solution, this gradient composition causes greater swelling at the top surface than the bottom to drive photocrease bending. Taken together, this suggests an origami-like shape-programming mechanism for photocreased MSPs, with backbone geometry dictated by pitch, angle, and degree of folding at the creases.

In addition to controlling the coiled filament geometry, spinning MSP filaments into bundles depends on coiling kinetics. The coiling speed of released MSPs was found to depend on the UV dose employed to produce the photocreases. For a series of copolymer 2 MSPs with $h_0$ -1 μm irradiated with $|\phi|$ -45° at $\lambda_{max} = 365$ nm, coiling speed increased with UV dose over experiments employing 25 (Supplementary Movie 3), 50, and 75 (Supplementary Movie 4) J cm⁻². This dose-dependent coiling rate suggests a route to more sophisticated geometries (e.g., knots) that demand precise spatiotemporal control (i.e., variable photocrease deformation rate along the length of a given MSP). No photocrease bending was observed for MSPs irradiated with ≤12.5 J cm⁻². This was consistent with behavior of copolymer 2 MSPs of comparable thickness that were subjected to heating without a preceding irradiation step and copolymer 3 MSPs (i.e., without PAG) that were irradiated (75 J cm⁻² and $\lambda_{max} = 365$ nm) and heated but exhibited negligible thickness transformation. In both cases, MSPs behaved as stiff elastic rods and were only induced to bend by introduction of local flow by pipette or capillary tube (Supplementary Movies 5 and 6). When the flow was stopped, the MSPs elastically recovered their original straight configuration, confirming that photocreases are required to achieve significant out-of-plane bending in MSPs of this thickness and -millimeter-scale length, and that photocrease formation requires mechanisms to chemically transform the polymers.

## Self-spinning, interlinked, twisted bundles

Chiral bundles were realized by printing MSP arrays at close proximity ($d = 50$–60 μm), embedding photocreases uniformly along the array, then dissolving the underlying sacrificial layer. Fig 4 shows frames from movies that capture the release and coiling process for MSP arrays where $\phi = 26°$ (Fig. 4a, Supplementary Movie 7) or 44° (Fig. 4b, Supplementary Movie 8), and $\tau$ - 3500$\Delta t$ (see Supplementary Notes 2). Each panel is labeled with the corresponding time in the bundling process ($t = 0$ s represents introduction of fluid). A latency period of

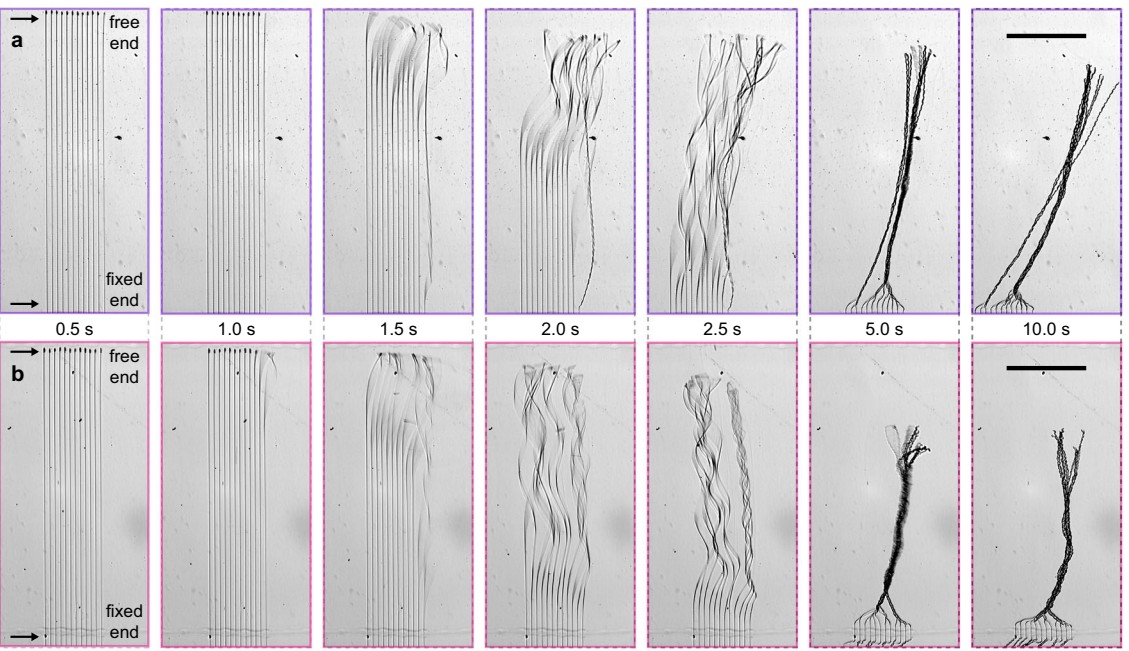

**Fig. 4 | Bundling kinetics.** Samples of aligned MSPs with $d = 60\,\mu m$ and $\phi =$ (**a**) 26° or (**b**) 44°. Micrographs show each sample at 0.5, 1.0, 1.5, 2.0, 2.5, 5.0 and 10.0 s after introduction of pH 8 buffer solution. Scale bars 1 mm.

~1.5 s was observed, likely corresponding to PSS dissolution (Fig. 4a, b left), during which the MSPs remained fixed to the substrate surface. By $t = 2$ s, most of the MSPs had detached from the substrate and begun coiling rapidly (Fig. 4a, b center), which continued for several seconds (2.5–5 s, Fig. 4a, b center-right right) until the helices achieved their final configuration (Fig. 4a, b far right). In Fig. 4a (26°), distinct initiation times were observed across the MSP array. At $t = 2$ s, most MSPs were visibly coiling; subsequent coil propagation (2.5 s) and equilibration (5 s) induced adjacent MSPs to spin together to afford two bundles comprising three and nine interlinked and helically programmed MSPs (10 s; note the bundle fixed end where individual MSPs remain adhered to the glass surface at the bottom of the micrograph). Fig 4b (44°) shows that the bundling process is robust across different photocrease tilt angles. Two initial aggregates formed at $t = 2.5$ s, which then wound together (5 s) to produce a single MSP bundle with four smaller branches toward the free end (10 s).

Figure 5a shows fluorescence micrographs of MSP arrays before release, while Fig. 5b–d shows 3D confocal z-stacks of the same arrays after release and bundling as a function of photocrease tilt angle, ribbons per bundle, and average as-printed thickness ($\phi$, $n$, $h_{o,avg}$) of (14°, 4, 0.93 μm) in Fig. 5b, (46°,7, 1.18 μm) in Fig. 5c, and (76°,10, 1.01 μm) in Fig. 5d. All of the bundles were left-handed, consistent with bottom-face-in coiling. The bundle in Fig. 5b ($\phi = 14°$) contains four MSPs in a left-handed helical configuration with a hollow center. The innermost and outermost MSPs show radii of ~20 μm and 24 μm, respectively, along the bundle length. This represents a ~2.5-fold increase in radius over a 'single' (i.e., unbundled) MSP of comparable $\phi$ and dimensions (Supplementary Fig. 11), suggesting that the MSPs are sterically disallowed from their equilibrium geometry and implying the application of a compressive force by each helix to the bundle. Interestingly, the linked bundles for the $\phi = 14°$ case (Fig. 5b, Supplementary Fig. 12a) showed negligible curvature along the main axis, suggesting relatively high bending energy that is consistent with the properties of single MSP helices with the same photocrease tilt angle. Bundles with greater curvature were observed for larger photocrease tilt angles (Supplementary Fig. 12b, c). The bundle in Fig. 5c (46°) is composed of seven left-handed helical MSPs twisted around a hollow core, with the radius of a constituent helix spanning from 20 μm

(innermost MSP) to 32 μm (exterior MSP). While fairly uniform, the helical paths traced by the MSPs in Fig. 5c exhibited more variability in their pitch and radius than those of Fig. 5b along the bundle length, with different MSPs exchanging layer positions at several points along the aggregate structure (Supplementary Fig. 13 center). The bundle in Fig. 5d (76°) is composed of 10 ribbons that trace a predominantly left-handed helix around a hollow core. In this case, the paths traced by the 10 constituent MSPs were highly variable, with a single MSP emanating from the main bundle (Fig. 5d left, dashed box) and dramatic variations in radial distance from the main axis for constituent filaments along the bundle length (Supplementary Fig. 13 right), reminiscent of 'fiber migration' well-known for macroscopically spun yarns[64]. Values for $p$ and $r$ (Supplementary Fig. 11) were estimated by determining the best fit helices from a short (60 μm length) segment of the bundle; the resulting radii spanned 18–55 μm.

For self-spun bundles, understanding the internal topology is essential for relating bundling mechanisms and structure to their properties. Bundle topologies for the three bundles in Fig. 5b–d were determined by connecting the centroids of constituent rigid segments via spline interpolation (Fig. 5e). The constituent MSPs for $\phi = 14°$ (Fig. 5e top) trace simple helices along the length of the bundle, with one MSP nestled inside another in what is effectively a 3-ply geometry. The MSPs with $\phi = 46°$ (Fig. 5e center) and $\phi = 76°$ (Fig. 5e bottom) adopted more complex paths. The extent of linking, or entanglement, was quantified in the bundles by using an extension of Gauss's linking number of open curves[53]. Critically, the total number of links per bundle increases dramatically from 9 (4-component bundle) to 270 (10-component bundle), confirming a strong dependence on $n$; the average number of pairwise links increased from 1.5 to 6. Moreover, these highly interlinked structures are composed of individual MSPs that are confined (even for the innermost MSPs) to larger $p$ and $r$ values than found in the single-helix case and can be described as bundles of strained elastic springs applying a compressive load to the rest of the bundle. This compression effect, combined with extensive interlinking, suggests that this bundling mechanism gives access to robust mesoscale structures that may readily withstand physical deformation. This was tested by subjecting substrate-bound bundles to cyclical fluid flow by repeated withdrawal and ejection through a capillary tube. The

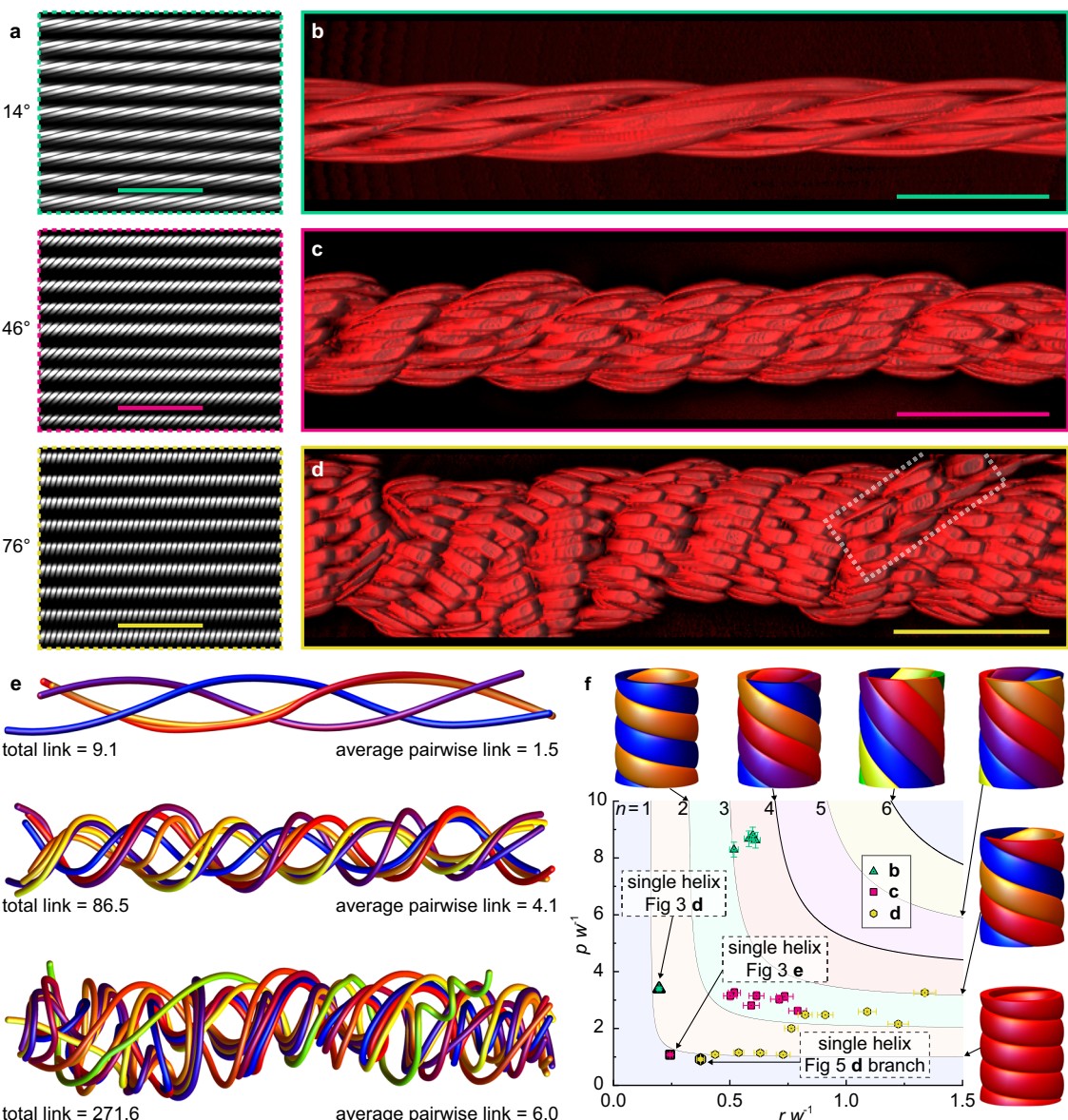

**Fig. 5 | Coiled MSP bundles. a** Fluorescence micrographs of substrate-bound, aligned MSP arrays with $\phi = 14°$ (top), 46° (center), and 76° (bottom); scale bars 200 μm. **b–d** 3D-reconstructed confocal z-stacks (right) of bundles arising from coincident and proximal MSP coiling with $(\phi,n) = (14°,4)$ (**b**), (46°,7) (**c**), and (76°,10) (**d**). The dashed box in **d** denotes a branch with $n = 1$; scale bars **b–d** 100 μm. **e** smoothed rendering of constituent MSP centerlines in bundles **b–d**; linking number increases with $n$. **f** Width-normalized pitch as function of width-normalized radius; the lower dimensional limit for an $n$-ply bundle is schematically indicated for $n = 1–6$ and is independent of helix chirality; data describes 'single' MSP helices and the constituent MSPs in bundles **b–d** indicating that constituent MSPs in **b–d** adopt dimensions of ideal 1, 2, or 3-ply structures. Error bars represent quadrature sum of rms displacement from the fitted helix (pitch, radius) with 1 standard deviation in measured MSP width.

bundles maintained their structural integrity with no visible deterioration or significant fraying, thus displaying a structural robustness even in turbulent flow with cyclical deformation. Moreover, the mechanics of 5-MSP bundles were compared to those of single helices by flow-induced bending. Specifically, MSPs at $\phi = 18, 34, 41, 54,$ and 60° were released into a pH 8 buffer solution containing 3 mM SDS to minimize interfacial energy and friction, affording a 5-MSP bundle and a single-MSP helix for each value of $\phi$ (Supplementary Fig. 20, Supplementary Movies 9–12, Supplementary Note 4). Bundles ($n = 5$) and helices ($n = 1$) were subjected to suction through a capillary tube at volumetric flow rates $Q$ (Supplementary Fig. 20a) to draw them across its inlet; above a critical flow rate $Q_c$, they bent in half and were drawn into the tube (Supplementary Fig. 20b, c). Notably, the $n = 5$ case proved robust when subjected to significantly higher flow rates than $n = 1$ before collapsing into the capillary tube, regardless of $\phi$. This

mechanical enhancement is demonstrated in Supplementary Movie 9, in which a bundle ($n = 5$ and $\phi = 18°$) withstands a flow of 500 μL min⁻¹ before springing back to its original shape. In contrast, in Supplementary Movie 10, an MSP helix ($n = 1$, $\phi = 18°$) immediately collapses at a lower flow rate of 300 μL min⁻¹. Supplementary Movies 11–12 show a similar enhancement for the $\phi = 60°$ case. Moreover, bundling increased $Q_c$ for all $\phi$ measured while decreasing $\phi$ dramatically increased the bending stiffness in both the $n = 1$ and $n = 5$ cases (Supplementary Fig. 20d and Supplementary Note 4).

In addition to mechanical enhancement, bundling significantly impacted the dimensions of the constituent coiled MSPs. Specifically, Fig. 5f compares bundles in Fig. 5b–d and single-MSP helices to ideal, closed-packed $n$-ply structures composed of helical (developable) ribbons, where $n = 1–6$ (depicted schematically)[26]. Contours describe the allowed packing structures of isometric spiral ribbons (i.e., the

lower limit allowed for ($r\,w^{-1}$, $p\,w^{-1}$)) for each value of $n$, represented visually by the surrounding schematics. The $n=1$ data lies near the lower bound predicted for a 1-ply helix, suggesting that photocrease bending in the single case continues until sterically prohibited. In all cases (Fig. 5b–d), the width-normalized $p$ and $r$ are larger for bundled MSPs than for $n=1$, consistent with the 'self-compressing bundle of springs' structure described above. The 4 MSPs of $\phi = 14°$ (triangular points and Fig. 5b) all occupy space accessible to a 3-ply bundle, and the 'interior' MSP is distinguishable from the three exterior MSPs due to its slightly smaller pitch and radius. Interestingly, as ($\phi$, $n$) increases to (46°, 7) (square points and Fig. 5c), helix dimensions decrease into the 2-ply region. A further increase to (76°,10) (hexagonal points and Fig. 5d) affords a larger spread of domains, with constituent MSPs showing variable $r$ but constant $p$ at the lower limit of a close-packed 1-ply helix. Subsequent radial layers are excluded from the bundle interior, with increasing $p$ and $r$ values until the outermost MSP enters the 3-ply regime.

## Discussion

Inspired by the ubiquity of bundled mesoscale filaments in Nature and the challenge of fabricating robust synthetic counterparts, we demonstrated a bottom-up assembly paradigm for self-spinning synthetic filaments into topologically linked bundles. At the scale of individual MSPs, embedded photocreased domains imparted local discretized control of filament curvature $\kappa$ and twist $\psi$ by locally embedding regions amenable to differential swelling, akin to origami-like mechanism for programming the 3D shapes of 1D ribbon-like structures. Importantly, this use of gradient swelling, a well-understood tool in shape-morphing materials, means that our design paradigm is readily generalizable beyond the chemistries described here. Repeated photocreases with the same tilt angle $\phi$ afford constant $\kappa$ and $\psi$ along the MSP length to generate achiral rods and rolls where $\phi = 0°$ and 90°, and helices of tunable pitch, radius, and chirality for all other values. When printed in close proximity and released in rapid succession by a flowing fluid, helically programmed MSPs twisted into linked bundles with tunable entanglement that hinged on the number $n$ of constituent MSPs. For example, a ~500 μm bundle segment with $n=10$ possessed more than 200 links, evidentially providing self-reinforcing mechanical integrity without physical attraction or covalent reinforcement between filaments. Importantly, the helical path of bundled MSPs reveals radius and pitch values far exceeding those observed for identical MSPs in the unbundled state, suggesting the presence of a compressive force applied by each MSP to the rest of the bundle. This is consistent with the observation that resulting linked bundles can be detached from the substrate and manipulated through solution without perceptible deterioration, confirming mechanical integrity. For individual MSPs, we predict that the locally encoded control of $\kappa$ and $\psi$ imparted by the photocrease platform will readily lend itself to the formation of arbitrary 3D paths via variable segment lengths, grayscale masks, and variable $\phi$. Similarly, our highly linked MSP bundles represent the first synthetic mesoscale correlate to natural fibrillar assemblies, and we anticipate that they will underpin an interesting and useful class of meso-to-macroscale collective structures.

## Methods
### Chemicals

Sodium 4-vinylbenzenesulfonate (Na4VBS, 90%), rhodamine B (95%), 4-dimethylaminopyridine (DMAP, 99%), N,N'-dicyclohexylcarbodiimide (DCC 99%), 2-hydroxyethyl methacrylate (HEMA 97% with 250 ppm monomethyl ether hydroquinone (MMEHQ) inhibitor), imidazole (99%), pH 1 buffer solution (glycine/sodium chloride/hydrogen chloride solution, Aldrich product number 1094321000), pH 8 buffer solution (boric acid/sodium hydroxide/hydrogen chloride solution, Aldrich product number 1094601000), pH 10 buffer solution

(boric acid/potassium chloride/sodium hydroxide solution, Aldrich product number 1094091000), alumina (activated, basic, Brockmann I), poly(sodium 4-styrenesulfonate) (PSS, $M_W$ 70 kDa, Aldrich product number 243051), sodium trifluoroacetate (98%), ethanol (absolute, 200-proof for molecular biology), sodium dodecyl sulfate (99%) (Aldrich), chloroform-d (CDCl₃, 99.8%), dimethylsulfoxide-d6 (DMSO-d6, 99.9%), and acetone-d6 (99.9%) (Cambridge Isotope Laboratories), 4-iodoaniline (99%), 4-bromo-1,8-naphthalic anhydride (95%) (TCI America), methanol (MeOH, Certified ACS, Fisher catalog number A412-20), dimethylformamide (DMF, Spectranalyzed™, Fisher catalog number D131-1), hexanes (Certified ACS, Fisher catalog number H292-20), isopropanol (IPA, Certified ACS, Fisher catalog number A416-20), chloroform (Certified ACS, 0.75% ethanol preservative, Fisher catalog number C298-20), diethyl ether (Certified ACS, BHT stabilized, Fisher catalog number E138-1), triphenylsulfonium chloride (TPSCl, 94%), toluene (Optima™, Fisher catalog number T291-4), anhydrous sodium sulfate (Certified ACS, 10-60 mesh, Fisher catalog number S415-212) (Fisher Scientific), lithium chloride (LiCl 99%), sodium thiosulfate (99%), trifluoromethanesulfonic acid (TfOH, 99%) (Acros Organics), and silica gel (Sorbent Technologies), were used as received without further purification. Triethylamine (TEA, 99.5%, Aldrich) and dichloromethane (DCM, stabilized, Certified ACS, Fisher catalog number D37-20, Fisher Scientific) were dried over calcium hydride (95%, Aldrich) and distilled. *Tert*-butyl methacrylate (TBMA, 98% with 200 ppm MMEHQ inhibitor, Aldrich) and glycidyl methacrylate (GMA, 97% with 100 ppm MMEHQ inhibitor, Aldrich) were purified by passage through a plug of basic alumina. 2,2'-Azobisisobutyronitrile (AIBN, 98%, Aldrich) was recrystallized from MeOH. *Meta*-chloroperbenzoic acid ($m$-CPBA, ≤ 77% purity, Aldrich) was dissolved to 50 mg mL⁻¹ in DCM and dried over sodium sulfate; activity was determined by iodometric titration with sodium iodide (99.9%, Alfa Aesar) and sodium thiosulfate. Tetrahydrofuran (THF, Optima™, Fisher catalog number T427-1 Fisher Scientific) was dried over sodium (Aldrich, 99.9%) benzophenone (99%, Aldrich) ketyl and distilled. Trifluoroethanol (TFE, 99.9%, Oakwood Chemical) was distilled. $N_2$ gas was dried by passing through a Drierite (W.A. Hammond Drierite Company) column.

### Instrumentation

¹H (500 MHz) and ¹⁹F (471 MHz) spectroscopic data were collected using a Bruker Ascend TM500 spectrometer with a Prodigy cryoprobe. ¹³C (100.6 MHz) was collected using a Bruker Avance NEO 400 MHz spectrometer with a 5 mm double resonance broad banded iProbe. **PAG2** was dissolved to 5 μg mL⁻¹ in methanol and measured in positive ion ESI mode (resolving power 60,000 at 202 m/z) using an Orbitrap Fusion mass spectrometer. Copolymer molecular weight was estimated against PMMA standards by gel permeation chromatography (GPC) at 1 mL min⁻¹ flow rate. Copolymers **1** and **3** were eluted in a mobile phase of 10 mM LiCl in DMF with an Agilent 1260 Infinity refractive index detector, Agilent 1260 Infinity isocratic pump, a 50 × 7.5 mm PL gel mixed guard column, a 300 × 7.5 mm PL gel 5 μm mixed C column, and a 300 × 7.5 mm PL gel 5 μm mixed D column at 50 °C. Copolymer **2** was eluted in a mobile phase of 20 mM sodium trifluoroacetate in trifluoroethanol using an Agilent 1200 series isocratic pump and refractive index detector, a 50 x 8 mm PFG guard column, and three 300 x 8 mm PFG analytical linear M columns (Polymer Standards Service). UV-ozone (UVO) surface treatment was conducted with a Jelight Company, Inc. Model 342 UVO-Cleaner®. Laser engraving was carried out using a Universal Laser Systems VLS3.50 laser engraver equipped with a 30 W CO₂ (10.6 μm) laser with 4.2% power, 40% speed, and 1000 ppi pulse rate. Flow-coating was carried out using a SmarAct, Inc SLC-1780s linear actuator. Near UV irradiation ($\lambda_{max}$ 365 nm) was performed with an OAI model 2105 500 W illumination controller or a Newport 97435 lamp housing with a Newport 69907 power supply and USHIO USH-508SA mercury arc lamp. A model 30 OAI Instruments 1000 Watt DUV Exposure System

equipped with a DUV 1000 lamp (Advanced Radiation Corporation) was used for all deep UV ($\lambda_{max}$ 254 nm) irradiation. Reactive Ion Etch (RIE) experiments employed an Advanced Vacuum Vision 320 MkII Reactive Ion Etch System with 50 sccm $O_2(g)$ flow rate, 50 mTorr chamber pressure, 100 W RF power, and 13.56 MHz RF frequency. Conventional optical microscopy was conducted on an Axio Observer 7 Materials microscope equipped with a Hamamatsu C11440 Orca-Flash4.0 Digital Camera, 2 Eppendorf TransferMan 4r micromanipulators connected to a New Era Pump Systems, Inc. NE-1000 syringe pump, an X-Cite 120LED (Excelitas Technologies), and Zeiss filter 45 for all copolymers. Confocal z-stacks were collected on a Nikon A1R Multiphoton Confocal Microscope with a 25x Nikon Apo LWD water-immersion lens, 325 nm z-step, 561 nm laser excitation, and 655 nm emission. Fourier-transform Infrared (FT-IR) data were collected in attenuated total reflectance mode using a PerkinElmer Spectrum One FT-IR Spectrometer equipped with a Universal ATR Sampling Accessory. Optical profilometry data was collected using a Zygo NewView 7300 Optical Surface Profiler. Capillary tubes (Chem-Glass, 1.0–1.1 mm O.D.) were used for flow measurements to assess single-helix and bundle flexibility.

### Synthesis of rhodamine B methacrylate (RBMA) monomer

The RBMA synthesis was also adapted from a reported procedure[40,65]. In brief, a 2-neck, 250 mL round-bottom flask with stir bar was flame-dried and purged with dry nitrogen gas, then rhodamine B (10 g, 20.9 mmol, 1 equivalent), DMAP (150 mg, 1.23 mmol, 0.06 equivalents), and DCC (5.2 g, 25.2 mmol, 1.21 equivalents) were added against positive flow of dry $N_2(g)$. The flask was sealed with a septum, then dry DCM (105 mL) and HEMA (3.1 mL, 25 mmol, 1.20 equivalents) were added by syringe. The solution was stirred at 20 °C for 25 h, then concentrated under reduced pressure and purified by column chromatography (silica gel stationary phase, 90:10 DCM:MeOH eluent) and dried under high vacuum to afford a dark purple powder (6.15 g, 50% yield). $^1$H NMR (500 MHz, CDCl3, δ) 8.33–8.26 (d, 1H, J = 7.90 Hz, aromatic), 7.88–7.81 (t, 1H, J = 7.5 Hz, aromatic), 7.79–7.72 (t, 1H, J = 7.7 Hz), 7.35–7.30 (d, 1H, J = 7.5 Hz), 7.10–7.03 (d, 2H, J = 9.5 Hz), 6.97–6.90 (dd, 2H, J = 9.5, 2.3 Hz), 6.82–6.77 (d, 2H, J = 2.2 Hz), 6.05–5.98 (s, 1H, vinyl), 5.58–5.52 (s, 1H, vinyl), 4.33–4.28 (t, 2H, J = 5.0 Hz, OC$H_2$C$H_2$O), 4.21–4.16 (t, 2H, J = 4.7 Hz, OC$H_2$C$H_2$O), 3.70–3.63 (8H, q, J = 7.2 Hz, NC$H_2$C$H_3$), 1.90–1.85 (s, 3H, methacrylate CC$H_3$), 1.37–1.29 (t, 12H, J = 7.05 Hz, NC$H_2$C$H_3$)

### Synthesis of triphenylsulfonium 4-vinylbenzenesulfonate PAG1

TPS-4-VBS was synthesized by adapting a procedure from a literature report[66]. In brief, 94% TPSCl (1.06 g, 3.33 mmol, 1 equivalent) and 90% Na4VBS (767 mg, 3.35 mmol, 1 equivalent) were combined and shaken with 3.3 mL RO water in a 20 mL scintillation vial to afford a brown emulsion. The brown organic phase was removed, and the aqueous phase extracted with 6 x 1 mL DCM. The combined organic phase was diluted to 12 mL, washed with 4 x 1 mL RO water, filtered to remove residual brown solid, concentrated, then diluted with hexanes (1 mL) to induce crystallization. Residual solvent was removed under reduced pressure to afford the desired product as white crystals (1.24 g, 83% yield). $^1$H NMR: (500 MHz, CDCl$_3$, δ): 7.86–7.81 (d, 2H, J = 8.23 Hz, 4-vinylbenzenesulfonate aromatic), 7.76–7.72 (d, 6H, J = 7.51 Hz, S$^+$(C$_6$H$_5$)$_3$), 7.70–7.66 (t, 3H, J = 7.42 Hz, S$^+$(C$_6$H$_5$)$_3$), 7.64–7.59 (t, 6H, J = 7.62 Hz, S$^+$(C$_6$H$_5$)$_3$), 7.30–7.27 (d, 2H, J = 8.20 Hz, 4-vinylbenzenesulfonate aromatic), 6.70–6.60 (dd, 1H, J = 10.89, 17.61 Hz, 4-vinylbenzenesulfonate vinyl), 5.74–5.65 (d, 1H, J = 17.61 Hz, 4-vinylbenzenesulfonate vinyl), 5.24–5.17 (d, 1H, J = 10.97 Hz, 4-vinylbenzenesulfonate vinyl).

### Synthesis of 4

This method was adapted from reported procedures[58,59]. 4-bromo-1,8-naphthalic anhydride (1.02 g, 3.68 mmol, 1.0 equivalent), 4-iodoaniline (1.61 g, 7.35 mmol, 2.0 equivalents), and imidazole (5.0 g, 73.4 mmol, 19.9 equivalents) were combined in a 100 mL round-bottom flask with 32 mL chloroform. The reaction vessel was fit with a reflux condenser and lowered into an oil bath at 85 °C to reflux for 3 h. Then, the reaction solution was cooled to 20 °C and the solvent removed under reduced pressure. The crude solid mixture was sonicated in 100 mL ethanol for 15 min and filtered to afford a white solid and purple supernatant solution. The solid was collected and rinsed with ethanol, then dried under reduced pressure at 40 °C to afford pure product **4** as a white solid (1.57 g, 89% yield). $^1$H NMR: (500 MHz, CDCl$_3$, δ): 8.73–8.68 (dd, 1H, naphthyl aromatic, BrCCCHCHC$\underline{H}$, J = 7.3, 1.0 Hz), 8.68–8.63 (dd, 1H naphthyl aromatic, BrCCCHC$\underline{H}$CH, J = 8.5, 1.0 Hz), 8.49–8.43 (d, 1H, 8.15–8.09, J = 7.9 Hz, naphthyl aromatic BrCC$\underline{H}$CH), 8.12–8.07 (d, 1H, J = 7.9 Hz, naphthyl aromatic BrCC$\underline{H}$CH), 7.93–7.85 (m, overlapping (1H, naphthyl aromatic BrCCCHC$\underline{H}$CH) and (2H, I(C$_2$H$_2$)(C$_2\underline{H}_2$)CN)), 7.10–7.03 (m, 2H, I(C$_2\underline{H}_2$)(C$_2$H$_2$)CN)).

### Synthesis of 5

This method was adapted from reported procedures[58,59]. TfOH (930 μL, 10.5 mmol, 5.0 equivalents), was added to a vigorously stirring mixture of **4** (1.00 g, 2.09 mmol, 1.0 equivalent), toluene (900 μL, 8.45 mmol, 4.0 equivalents), m-CPBA (44 mL of 202 mM solution in DCM, 8.89 mmol, 4.0 equivalents), and DCM (14 mL) in a 250 mL round-bottom flask at 20 °C. The solution immediately turned brown and heat evolved. The reaction was quenched after 2.5 h by adding 125 mL diethyl ether, then the precipitated solids were isolated by filtration and rinsed with diethyl ether. Residual solvent was removed under reduced pressure to afford iodonium triflate salt **5** (1.21 g, 81% yield). $^1$H NMR: (500 MHz, acetone-d$_6$, δ): 8.71–8.67 (d, 1H, J = 8.5 Hz, naphthyl aromatic BrCCCHCHC$\underline{H}$), 8.67–8.62 (m, 1H, naphthyl aromatic BrCCC$\underline{H}$CHCH), 8.53-8.47 (d, 2H, J = 8.7 Hz, I$^+$(C$_2$H$_2$)(C$_2$H$_2$)CN), 8.45–8.40 (d, 1H, J = 7.9 Hz, naphthyl aromatic BrCC$\underline{H}$CH), 8.36–8.31 (d, 2H, J = 8.4 Hz, I$^+$(C$_2$H$_2$)(C$_2$H$_2$)CCH$_3$), 8.28–8.24 (d, 1H, J = 7.9 Hz, naphthyl aromatic BrCC$\underline{H}$CH), 8.09-8.03 (t, 1H, J = 7.9 Hz, naphthyl aromatic BrCCCHC$\underline{H}$CH), 7.75-7.69 (d, 2H, J = 8.7 Hz, I$^+$(C$_2$H$_2$)(C$_2$H$_2$)CN), 7.50–7.45 (d, 2H, J = 8.2 Hz, I$^+$(C$_2$H$_2$)(C$_2$H$_2$)CCH$_3$), 2.47–2.43 (s, 3H, I$^+$(C$_2$H$_2$)(C$_2$H$_2$)CCH$_3$). $^{13}$C NMR: (100.6 MHz, DMSO-d$_6$, δ): 162.99 (s, 1C, naphthalimide carbonyl), 162.92 (s, 1C, naphthalimide carbonyl), 142.93 (s, 1C, aromatic), 139.14 (s, 1C, aromatic), 135.87 (s, 2C, aromatic), 135.42 (s, 2C, aromatic), 132.98 (s, 1C, aromatic), 132.59 (s, 2C, aromatic), 132.57 (s, 2C, aromatic), 131.70 (s, 1C, aromatic), 131.41 (s, 1C, aromatic), 131.02 (s, 1C, aromatic), 129.82 (s, 1C, aromatic), 129.57 (s, 1C, aromatic), 128.82 (s, 1C, aromatic), 128.60 (s, 1C, aromatic), 122.98 (s, 1C, aromatic), 122.18 (s, 1C, aromatic), 116.20 (s, 1C, aromatic), (122.56, 122.36, 119.15, 115.95, q, 1C, triflate carbon, split by fluoro groups), 113.02 (s, 1C, aromatic), 20.97 (s, 1C, C$\underline{H}_3$). $^{19}$F NMR: (471 MHz, acetone-d$_6$, δ): 78.83–78.85 (s, triflate).

### Synthesis of diaryliodonium 4-vinylbenzenesulfonate PAG2

A solution of **5** (1.08 g, 1.51 mmol, 1.0 equivalent) in 220 mL DCM was washed with 100 mL of 131 mM Na4VBS (14.5 mmol, 9.6 equivalents) in a 1 L separatory funnel. Then, the organic phase was separated and the aqueous phase extracted with 50 mL DCM. Both organic phases were combined, then this process was repeated with 2 x 100 mL of 131 mM Na4VBS solution. The resulting organic phase was dried over sodium sulfate, then decanted and the drying agent rinsed with an additional 5 x 50 mL DCM. The organic phases were combined and concentrated under a stream of air, hexanes were added to facilitate precipitation, then solvent was removed under reduced pressure to afford a solid powder that was crushed by mortar and pestle before removing trace solvents under vacuum to afford **PAG2** as a solid powder with estimated 3 wt% residual Na4VBS (1.17 g, 100% yield). See Supplementary Fig. 21 for schematic depiction of synthetic pathway. $^1$H NMR: (500 MHz, DMSO-d$_6$, δ): 8.67–8.61 (d, 1H, J = 8.5 Hz, naphthyl aromatic BrCCCHCHC$\underline{H}$),

8.60–8.55 (d, 1H, J = 7.2 Hz, naphthyl aromatic BrCCC<u>H</u>CHCH), 8.42–8.36 (d, 2H, J = 8.5 Hz, I⁺(C₂<u>H</u>₂)(C₂<u>H</u>₂)CN), 8.36–8.31 (d, 1H, J = 7.9 Hz, BrCC<u>H</u>CH), 8.31–8.25 (d, 1H, J = 7.9 Hz, BrCC<u>H</u>CH), 8.25-8.20 (d, 2H, J = 8.2 Hz, I⁺(C₂<u>H</u>₂)(C₂<u>H</u>₂)CCH₃), 8.08–8.01 (t, 1H, J = 7.9 Hz, naphthyl aromatic BrCCC<u>H</u>CHCH), 7.61–7.53 (m, 4H, overlapping (2H, I⁺(C₂<u>H</u>₂)(C₂<u>H</u>₂)CN) and (2H, 4-vinylbenzenesulfonate aromatic)), 7.44–7.36 (m, 4H, overlapping (2H, I⁺(C₂<u>H</u>₂)(C₂<u>H</u>₂)CCH₃) and (2H, 4-vinylbenzenesulfonate aromatic)), 6.79-6.65 (dd, 1H, J = 17.7, 10.9 Hz, 4-vinylbenzenesulfonate vinyl), 5.89–5.78 (d, 1H, J = 17.1 Hz, 4-vinylbenzenesulfonate vinyl), 5.31–5.21 (d, 1H, J = 10.9 Hz, 4-vinylbenzenesulfonate vinyl), 2.42–2.31 (s, 3H, I⁺(C₂<u>H</u>₂)(C₂<u>H</u>₂)CC<u>H</u>₃). ¹³C NMR: (100.6 MHz, DMSO-d₆, δ): 162.99 (s, 1C, naphthalimide carbonyl), 162.92 (s, 1C, naphthalimide carbonyl), 147.69 (s, 1C, 4-vinylbenzenesulfonate aromatic ⁻O₃S<u>C</u>), 142.74 (s, 1C, iodonium aromatic), 139.00 (s, 1C, iodonium aromatic), 137.22 (s, 1C, 4-vinylbenzenesulfonate aromatic CH₂-CH-<u>C</u>),136.22 (s, 1C, 4-vinylbenzenesulfonate vinyl CH₂-CH-C), 135.88 (s, 2C, iodonium aromatic), 135.44 (s, 2C, iodonium aromatic), 132.98 (s, 1C, iodonium aromatic), 132.51 (s, 2C, iodonium aromatic), 132.48 (s, 2C, iodonium aromatic), 131.70 (s, 1C, iodonium aromatic), 131.43 (s, 1C, iodonium aromatic), 131.05 (s, 1C, iodonium aromatic),129.81 (s, 1C, iodonium aromatic), 129.58 (s, 1C, iodonium aromatic), 128.85 (s, 1C, iodonium aromatic), 128.60 (s, 1C, iodonium aromatic), 125.90 (s, 2C, 4-vinylbenzenesulfonate aromatic), 125.54 (s, 2C, 4-vinylbenzenesulfonate aromatic), 122.95 (s, 1C, iodonium aromatic), 122.16 (s, 1C, aromatic), 116.47 (s, 1C, iodonium aromatic), 114.83 (s, 1C, 4-vinylbenzenesulfonate vinyl <u>C</u>H₂-CH-C), 113.32 (s, 1C, iodonium aromatic), 21.00 (s, 1C, iodonium aliphatic <u>C</u>H₃). ESI (m/z): 567.94 (C₂₅H₁₆BrINO₂⁺, calculated: 567.94, Supplementary Fig. 23).

## Synthesis of copolymer 1

TBMA (2.3 mL, 14 mmol, 89 equivalents), TPS-4-VBS (200 mg, 0.45 mmol, 2.8 equivalents), GMA (39 μL, 0.29 mmol, 1.9 equivalents), RBMA (20 mg, 34 μmol, 0.2 equivalents), and AIBN (26 mg, 0.16 mmol, 1 equivalent) were dissolved in DMF (5 mL) in a 20 mL scintillation vial equipped with a stir bar, then degassed by bubbling for 30 min with dry N₂(g) while stirring at 20 °C. After degassing, the septum was covered with a piece of electrical tape and the vial was transferred to an aluminum block, where the mixture was stirred at 80 °C for 22 h. The reaction was stopped by cooling to −20 °C, then purified by precipitating into 65:35 water:MeOH, re-dissolving in THF, precipitating three times in stirring hexanes, and finally drying under high vacuum at 20 °C for 18 h to yield the desired product. (1.03 g, 45%) as a pink powder. ¹H NMR: (500.13 MHz, CDCl₃, δ): 7.88–7.80 (d, 6H, J = 7.69 Hz, S⁺(C₆<u>H</u>₅)₃), 7.79–7.71 (br s, 2H, 4-vinylbenzenesulfonate aromatic), 7.74–7.69 (t, 3H, J = 7.39 Hz, S⁺(C₆<u>H</u>₅)₃), 7.69–7.61 (t, 6H, J = 7.64 Hz, S⁺(C₆<u>H</u>₅)₃), 7.10-6.93 (br s, 2H, 4-vinylbenzenesulfonate aromatic), 4.37-4.03 (br m, overlapping (1H, GMA COOC<u>H</u>H)[67] and (4H, RBMA OC<u>H</u>₂C<u>H</u>₂O), 3.97–3.78 (br s, 1H, GMA COOCH<u>H</u>)[67], 3.70–3.57 (br m, 8H, RBMA (N(C<u>H</u>₂CH₃)₂)₂), 3.27–3.13 (br s, 1H, GMA COOCH₂C<u>H</u>OCHH)[67], 2.91–0.14 (br m, mixed aliphatic), 2.86-2.77 (br s, 1H, GMA COOCH₂CHOC<u>H</u>H)[67], 2.69–2.57 (br s, 1H, GMA COOCH₂CHOCH<u>H</u>)[67], 1.50–1.35 (br m, 9H, TBMA C(C<u>H</u>₃)₃) (Supplementary Fig. 14). GPC: (DMF with 10 mM LiBr, PMMA standards): $M_n$ = 21 kDa, $M_w$ = 46 kDa, Đ = 2.2 (Supplementary Fig. 17).

## Synthesis of copolymer 2

RBMA (146 μL of 50 mg mL⁻¹ solution in DMF, 12.4 μmol, 0.1 equivalents), **PAG2** (450.4 mg, 0.60 mmol, 4.8 equivalents), and DMF (5.55 mL) were combined in a 20 mL scintillation vial and stirred until the solution cleared. Then, GMA (31 μL, 0.23 mmol, 1.9 equivalents) and TBMA (2 mL, 12.3 mmol, 100 equivalents) were added; the solution initially clouded but cleared again with stirring. 150 μL of 50 mg mL⁻¹ AIBN (66.1 μmol, 0.5 equivalents) and the vial

was sealed with a septum. The solution was degassed by bubbling with dry N₂(g) for 30 min, then the reaction vessel was placed in an aluminum block at 80 °C and stirred for 19 h. The reaction was stopped by cooling to −20 °C, then precipitated into 400 mL of 80:20 MeOH:water. The polymer was collected by filtration and ground to a fine powder in fresh 80:20 MeOH:water. The resulting pink slurry was centrifuged at 5000 rpm for 15 min and the supernatant decanted, then the solid product was heated to 80 °C under reduced pressure for 20 min. The polymer was thrice re-dissolved to 12 mL in acetone, precipitated into 250 mL stirring hexanes, filtered, crushed via mortar and pestle, and heated to 80 °C under reduced pressure to afford 1.23 g (55% yield) of copolymer 2 as a pink powder. ¹H NMR: (500.13 MHz, CDCl3, δ): 8.71–8.53 (s, 2H, naphthyl aromatic), 8.45–8.31 (s, 1H, naphthyl aromatic), 8.21–8.09 (s, 2H, I⁺(C₂<u>H</u>₂)(C₂H₂)CN), 8.09–8.02 (s, 1H, naphthyl aromatic BrCC<u>H</u>), 8.01–7.92 (s, 2H, I⁺(C₂<u>H</u>₂)(C₂H₂)CCH₃), 7.92–7.81 (s, 1H, naphthyl aromatic), 7.72–7.42 (br s, 2H, 4-vinylbenzenesulfonate aromatic), 7.42–7.32 (s, 2H, I⁺(C₂<u>H</u>₂)(C₂<u>H</u>₂)CN) overlapping with solvent residual), 7.32–7.16 (s, 2H, I⁺(C₂<u>H</u>₂)(C₂<u>H</u>₂)CCH₃) overlapping with solvent residual), 7.14–6.97 (br s, 2H, 4-vinylbenzenesulfonate aromatic), 4.44–4.03 (br m, overlapping (1H, GMA COOC<u>H</u>H)[67], and (4H, RBMA OC<u>H</u>₂C<u>H</u>₂O), 4.04–3.76 (br s, 1H, GMA COOCH<u>H</u>)[67], 3.74–3.55 (br m, 8H, RBMA (N(C<u>H</u>₂CH₃)₂)₂), 3.29–3.08 (br s, 1H, GMA COOCH₂C<u>H</u>OCHH)[67], 2.88–2.75 (br s, 1H, GMA COOCH₂CHOC<u>H</u>H)[67], 2.69–2.52 (br s, 1H, GMA COOCH₂CHOCH<u>H</u>)[67], 2.50–2.33 (br s, 3H, I⁺(C₂<u>H</u>₂)(C₂<u>H</u>₂)CC<u>H</u>₃), 2.29–0.27 (br m, mixed aliphatic), 1.50–1.35 (br m, 9H, TBMA C(C<u>H</u>₃)₃) (Supplementary Fig. 15). GPC: (TFE with 20 mM sodium trifluoroacetate, PMMA standards): $M_n$ = 21 kDa, $M_w$ = 34 kDa, Đ = 1.7 (Supplementary Fig. 18).

## Synthesis of copolymer 3

TBMA (5.7 mL, 35 mmol, 349 equivalents), RBMA (50 mg, 85 μmol, 0.85 equivalents), and AIBN (16.5 mg, 100 μmol, 1 equivalent) were dissolved in 17.6 mL THF in a 50 mL round-bottom flask that was sealed with a septum and degassed by bubbling with dry N₂ (g) for 30 min. The reaction vessel was transferred to a 60 °C oil bath to stir for 23 h, then precipitated into 500 mL of 90/10 MeOH water and dried under reduced pressure to afford 3.84 g (76% yield) copolymer 3 as a pink powder. ¹H NMR: (500.13 MHz, CDCl₃, δ): 2.20–0.78 (br m, mixed aliphatic), 1.50–1.35 (br m, 9H, TBMA C(C<u>H</u>₃)₃) (Supplementary Fig. 17). GPC: (DMF with 10 mM LiBr, PMMA standards): $M_n$ = 21 kDa, $M_w$ = 35 kDa, Đ = 1.7 (Supplementary Fig. 19).

## Characterization of copolymer photoactivity

Copolymer 1 was dissolved to 100 mg mL⁻¹ in toluene, drop-cast (5 μL) onto glass slides, and allowed to dry without heating. Then, the films were characterized by ATR IR i) without further processing, ii) after heating to 150 °C for 60 s; and iii) after irradiating at λ = 254 nm for a dose of 900 mJ cm⁻², then heating to 150 °C for 60 s. The change in thickness resulting from cleavage of t-butyl esters during photocrease formation (copolymers 1 and 2) was measured by optical profilometry.

## Substrate preparation, flow-coating, and release

Glass slides (24 x 40 x 0.17 mm³, Fisher Scientific) were cleaned by sonication for 15 min each in soapy water, reverse osmosis water, and isopropanol, followed by 15 min of surface treatment by UV-ozone to render the surface hydrophilic. Then, a 2 wt% solution of PSS in RO water was applied by spin-coating onto the hydrophilized glass surface (10 s at 500 RPM, then 40 s at 2000 RPM). PSS-coated slides were laser engraved (4.2% power, 40% speed, 1000 PPI) at 1–38 mm intervals to afford stripes of bare glass to which MS(BC)Ps would adhere upon flow-coating and release. Then, the substrates were fixed to a translating stage, and a razor blade taped to a stationary mount was lowered to a height of ~ 200 μm above the substrate surface. A polymer-in-toluene solution (4–7 μL of 4–5 mg mL⁻¹

polymer) was injected between the blade and substrate to afford a capillary bridge 24–38 mm in length. The substrate was translated in 40-400 µm intervals at 3 mm s⁻¹ with a 3-18 s delay between steps to deposit the MSPs, which were irradiated through a photomask at (i) 200–1000 mJ cm⁻² at $\lambda = 254$ nm (copolymer **1**), (ii) 0–75,000 mJ cm⁻² at $\lambda = 365$ nm or 5000 mJ cm⁻² at $\lambda = 254$ nm (copolymer **2**), or (iii) 75,000 mJ cm⁻² at $\lambda = 365$ nm (copolymer **3**), and heated to 150 °C for 60–120 s to afford photocreased MSPs (copolymers **1** and **2**). MSPs were cut into 1–38 mm segments *via* laser engraver and subjected to reactive ion etching with $O_2$ plasma for 30 s to remove any residual inter-MSP polymer film. To release MSPs, two strategies were employed, depending on the intended experiments. For samples using micromanipulators or syringe pump to manipulate MSP helices or bundles, an aqueous solution was prepared by filling a polystyrene Petri dish (Fisher Scientific, 60 mm diameter, 15 mm depth) with 10 mL of pH 1, 8, or 10 buffer solution either as-received or with 3 mM dissolved SDS. Then, a coated substrate was gently floated on top of the solution and quickly submerged using tweezers. Upon submersion, the underlying PSS layer dissolved to release the MSPs. Glass capillary tubes were inserted into a Capillary Holder 4 (Eppendorf), which was mounted in a TransferMan 4r micromanipulator (Eppendorf) and connected to a syringe pump loaded with aqueous solution for flow experiments. Thus equipped, the end of capillary tube was lowered into the aqueous solution to enable hand-controlled manipulation of MSP helices and bundles. For confocal and release kinetics experiments, an enclosed chamber with an inlet and outlet was constructed around the MSP-containing glass surface by using 2 additional glass slides (24 x 40 x 0.17 mm³) as spacers to separate the MSP-coated surface from a 75 x 25 x 1 mm³ slide, with Loctite 5-min quick-set epoxy as glue. Epoxy was allowed to cure for at least 4 h, then the chamber was flushed with aqueous solution (RO water or pH 1, 8, or 10 buffer solution) to release the MSPs. Coiling was tracked in real time by conventional optical microscope, then the chamber was sealed with vacuum grease and parafilm (to prevent leaking or evaporation of the continuous phase) and characterized by laser scanning confocal microscopy to visualize 3D structure.

### Extracting segment centroid positions from single helices

Single (i.e., non-bundled) photocreased MSP helices were characterized by determining the centroid of each MSP segment using 3D segmentation, 3D binary processing, and 3D measurement tools in NIS Elements general analysis 3 toolbar to threshold the raw 3D data, filter segments by volume, color segments by ID, and calculate segment centroids. The primary axis of each data set was determined in matlab by finding its covariance matrix and the data was reorganized in order of position along the principal axis. Then, each data set was input into HELFIT[60,61] to determine the pitch and radius of the helix of best fit. Radial uncertainty was determined by the root mean square (rms) radial displacement of each centroid from the cylinder defined by the helix of best fit. Then, radial uncertainty was subtracted in quadrature from the rms displacement of each centroid from the best fit helix to determine pitch uncertainty. 4 of the 55 single-helix data sets used in this study afforded erroneous outputs via HELFIT analysis: these were divided into smaller sequences of 8–20 centroids and input to HELFIT. For these data sets, the resulting values were averaged; uncertainties for each sequence were calculated as described above, added in quadrature, and added to the standard deviation for the combined pitch and radius outputs. A sequence with erroneous output ($r = 0$) was omitted from average and uncertainty calculations for the associated data point.

### Extracting segment centroid positions from bundled MSPs

Segment centroids for bundled ribbons were determined by displaying the 3D data for each bundle in the 3D viewer ImageJ plugin and scanning along the x axis. The maximum and minimum x value for each segment were averaged to determine the x centroid position; y and z positions of each centroid were determined by manually selecting the yz pixel location within the chosen x slice. The individual MSPs in the bundles of Fig. 5b, c formed sufficiently uniform helices that each MSP was input into HELFIT as a complete data set to determine in-bundle pitch and radius. The MSPs of Fig. 5d traced nonuniform paths through the bundle; truncated segments for x spanning 215–275 µm were input to HELFIT. The smoothed MSP traces shown in Fig. 5e are spline interpolations of 3D segment centroid positions.

### Link calculation

Linking calculation of the open MSP curves was adapted from reported methods[53,68]; MSP ends were extrapolated until the value of the Gauss linking integral for each MSP pair converged. Average pairwise link denotes the average number of entanglements per MSP pair in each bundle; total link is the sum of links across all MSP pairs in the bundle.

### Contour plot determination for *n*-ply helical bundle

The contours plotted in Fig. 5f represent the lower bound of *p* and *r* when *n* ribbons of width *w* are twisted around the same central axis to afford *n*-ply helical bundles[26] of helix angle *α* confined to an identical radius. This lower bound is established as follows. First, we consider that the lower limit of *p* is established in the case of self-contact ($n = 1$) or inter-ribbon contact ($n > 1$) as a function of *w*, *n*, and *α* according to (1), while *α* relates *p* and *r* according to (2).

$$p \geq \frac{w\,n}{\cos(\alpha)} \tag{1}$$

$$\cos(\alpha) = \frac{2\pi r}{\sqrt{p^2 + (2\pi r)^2}} \tag{2}$$

Then, (2) is substituted into (1) to describe the lower limit of *p* in terms of *w*, *n*, *p*, and *r*.

$$p \geq \frac{w\,n\,\sqrt{p^2 + (2\pi r)^2}}{2\pi r} \tag{3}$$

Finally, (3) is rearranged to isolate $p\,w^{-1}$, then *r* (right-hand-side numerator and denominator) is width-normalized ($(w^{-1}) / (w^{-1})$) to afford (4).

$$\frac{p}{w} \geq \frac{2\pi\left(\frac{r}{w}\right)}{\sqrt{\left(\frac{2\pi}{n}\left(\frac{r}{w}\right)\right)^2 - 1}} \tag{4}$$

Supplementary Fig. 22 provides a visual counterpart to this derivation.

### Determination of bundle central axis for radial migration plotting

The central axis direction of each bundle was estimated by visually aligning each data set to view through the empty bundle core. Then, all data points were projected onto the orthogonal plane and the average projected position used as a central axis intercept. The distance between each segment centroid and the central axis was plotted as a function of position along this central axis to afford the data sets reported in Supplementary Fig. 13. The plotted data and central axis determination for the 10-ribbon bundle of Fig. 5d omitted data for x < 175 µm.

## Data availability

The source data generated in this study have been deposited in the ScholarWorks at UMass Amherst[69] database under accession code

https://scholarworks.umass.edu/data/159 and https://doi.org/10.7275/6ybn-g003.

## Code availability

The source code generated in this study have been deposited in the ScholarWorks at UMass Amherst[69] database under accession code https://scholarworks.umass.edu/data/159 and https://doi.org/10.7275/6ybn-g003.

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

## Acknowledgements

Mass spectral data were obtained at the University of Massachusetts Amherst Mass Spectrometry Center. Confocal Microscopy data were gathered in the Light Microscopy Facility and Nikon Center of Excellence at the Institute for Applied Life Sciences at the University of Massachusetts, Amherst D.M.B. acknowledges an NDSEG Fellowship. G.M.G. acknowledges support from the US. National Science Foundation through Grant No. NSF DMR 2028885. A.J.C. acknowledges support by, or in part by, the U. S. Army Research Laboratory and the U. S. Army Research Office under contract/grant number W911NF-15-1-0358.

## Author contributions

D.M.B., G.M.G, A.J.C., and T.E. designed the study. D.M.B. conducted the synthesis and characterization of small molecules and polymers. D.M.B. conducted experiments on the fabrication and assembly of mesoscale polymer ribbons. D.M.B. conducted optical micrscopy and confocal microscopy experiments to characterize the mesoscale polymer ribbons and their assemblies. D.M.B. wrote code and generated the 3D visualization of the centroid data. G.M.G. wrote code to generate fits to centroid data in order to map 3D paths of bundled mesoscale polymer ribbons, and G.M.G. generated radial migration data. A.J.C., G.M.G., and T.E. supervised the project and commented on the data. D.M.B. and A.J.C. curated the data. D.M.B. prepared the manuscript with input from G.M.G., A.J.C., and T.E. All authors participated in the revisions of the manuscript and have confirmed the final version of the manuscript.

## Competing interests

The authors declare no competing interests.
