## [Peer Review File · Nature Communications]

Self-Spinning Filaments for Autonomously Linked MicrofibersREVIEWER COMMENTS

Reviewer #2 (Remarks to the Author):

This work by Barber et al. describes a bottom-up approach for the self-spinning of chiral mesoscale filaments using block copolymers into robust bundled structures that mimic natural fibrillar assemblies. The authors design mesoscale polymer (MSP) filaments using block copolymers embedded with a t-butyl methacrylate functionality that is sensitive to light, glycidyl methacrylate to promote cross-linking and triphenylsulfonium or naphthalimide-substituted diaryliodonium as photoacid generators enabling self-spinning of the filaments. Additionally, the block copolymers are functionalized with rhodamine B-substituted methacrylate in-order to enable easy visualization of the filaments by fluorescence microscopy. MSPs are deposited onto a glass substrate using a flow-coating process. Exposure to UV light forms an array of photoacreses. When printed in proximity and released in rapid succession by flowing-fluid, the MSP's twisted into linked bundles with tunable entanglement. It was found that bundling occurs when the inter-MSP spacing was small enough and the initiation time interval (measured when the MSPs are released from the substrates) was shorter than the coiling timescale. The tilt angle, including the handedness, of the filaments determines their 3D assembly. The mechanical integrity of the filaments was confirmed by removing the linked bundles from the substrate and exposing them to a buffered solution where no perceptible deterioration was observed.

The main innovation of this manuscript is in the statement: "To this end, we introduce mesoscale polymer (MSP) filaments with programmed local (i.e., arc-length-dependent) curvature and twist such that, upon application of a stimulus, each filament deforms to trace a predetermined 3D path in space, an accomplishment that has been confined to the macroscale before this work." The manuscript is well-written, and the research conducted herein fills an important gap in the field of biomaterials.

Minor comments:

- Please include the GPC traces and ¹H-NMR spectra for the block copolymers synthesized.
- In Fig. 2a, please include the structure of rhodamine in the block copolymer for easier reading.
- Please include the x, y, and z values for the block copolymer in Fig. 2a and discuss the rationale behind the ratio chosen for each block.
- Are the polymers random (at least looking at the experimental section that seems to be the case)? Please provide information on polymerization kinetics. Will the nature of the polymer (i.e., random vs. block vs statistical) impact the fibrillar self-assembly upon UV exposure?
- Multiple copolymers with different photoacid generators are reported. The manuscript, however, does not discuss the fibrillar assemblies of 1 neither does it provide a comparison between the copolymers. The rationale for using two photoacid generators is unclear. Please discuss the significance and assembly behavior of copolymer 1.
- Fig 4 caption, please move the (a) at the beginning of the caption.
- Please include page numbers for citation 63.
- The ratio of monomers for the three copolymers in synthetic procedure are different from the table in Fig S1.
- The authors investigated the effect of thickness h_0 on the dimension of the formed helices. How about the width of the alternating domains? Is there a strict requirement such as a specific value of a ratio for the width of the irradiated domain and masked domain? As far as I am concerned, this would play a bigger role than thickness on determining the helical pitch

and radius.

- The helicity of Fig 5f is right-handed while the helices in Fig 5a-e are left-handed. Is this a mistake?
- Why is the dispersity for copolymer 1 so much broader than 2 and 3?

Reviewer #3 (Remarks to the Author):

The manuscript reports on, allegedly, the first synthetic mesoscale fiber/ply/rope assembly. This is a synthetic equivalent to the numerous natural examples of fiber assemblies (collagen, keratin, and many others).

A lot is said in the manuscript on how to achieve these fiber bundles/ropes and the geometrical properties of the obtained structures. Few is told about their mechanical properties. Being a mechanical engineer, I am much more interested in the later (mech. prop.) than in the chemical methods used to obtain these ropes.

For the reader from outside Polymer Chemistry, the main query that arises on the fabrication scheme is how symmetry breaking is achieved: how the photocreasing and the dynamics of the swelling eventually produce 'bottom-face-in' bending and not the other way around. I think I understood it, but would not bet my salary that I got it right!

After all, one wants to understand what is the mechanical force that drives the helix formation.

The reader is left with the question: are these synthetic ropes mechanically better than natural fibril assemblies, or are they chemically more stable, less prone to aging or acid attacks, or...

remarks:

(a) authors might want to insist on the fact that all this is done in an aqueous solution. The average reader outside the field of Polymer Chemistry might not get this at first.

(b) Figure 3b. I get the feeling that a better choice for the axes would make the plot more interesting/universal. Why not plot pitch times $\tan \phi$ against radius + thickness for example?

REVIEWER COMMENTS

Reviewer #2 (Remarks to the Author):

This work by Barber et al. describes a bottom-up approach for the self-spinning of chiral mesoscale
filaments using block copolymers into robust bundled structures that mimic natural fibrillar assemblies. The
authors design mesoscale polymer (MSP) filaments using block copolymers embedded with a t-butyl
methacrylate functionality that is sensitive to light, glycidyl methacrylate to promote cross-linking and
triphenylsulfonium or naphthalimide-substituted diaryliodonium as photoacid generators enabling self-
spinning of the filaments. Additionally, the block copolymers are functionalized with rhodamine B-
substituted methacrylate in-order to enable easy visualization of the filaments by fluorescence microscopy.
MSPs are deposited onto a glass substrate using a flow-coating process. Exposure to UV light forms an
array of photocreases. When printed in proximity and released in rapid succession by flowing-fluid, the
MSP's twisted into linked bundles with tunable entanglement. It was found that bundling occurs when the
inter-MSP spacing was small enough and the initiation time interval (measured when the MSPs are
released from the substrates) was shorter than the coiling timescale. The tilt angle, including the
handedness, of the filaments determines their 3D assembly. The mechanical integrity of the filaments was
confirmed by removing the linked bundles from the substrate and exposing them to a buffered solution
where no perceptible deterioration was observed.

The main innovation of this manuscript is in the statement: "To this end, we introduce mesoscale polymer
(MSP) filaments with programmed local (i.e., arc-length-dependent) curvature and twist such that, upon
application of a stimulus, each filament deforms to trace a predetermined 3D path in space, an
accomplishment that has been confined to the macroscale before this work." The manuscript is well-written,
and the research conducted herein fills an important gap in the field of biomaterials.

Minor comments:

- • Please include the GPC traces and 1H-NMR spectra for the block copolymers synthesized.
○ See new Figures S14-16 for ¹H NMR spectra of copolymers 1-3 and S17-19 for GPC traces
of copolymers 1-3
- • In Fig. 2a, please include the structure of rhodamine in the block copolymer for easier reading.
○ Figure 2a has been updated as requested.
- • Please include the x, y, and z values for the block copolymer in Fig. 2a and discuss the rationale
behind the ratio chosen for each block.
○ This information is now given in Fig. 2a.
○ In this work, we have prepared and utilized random copolymers rather than block
copolymers (see response to the next comment for a more detailed explanation), which is
noted in the text. These random copolymers contain a high content of t-butyl ester groups,
which is advantageous for maximizing contrast in polarity between the masked
(hydrophobic) segments and the irradiated (swellable and hydrophilic) photocreases. The
PAG-substituted monomer units allow for direct inclusion of photoacid generators into the
polymer backbone, while the rhodamine B-substituted units afford photoluminescence and
the glycidyl methacrylates affect crosslinking. Numerous experiments led to optimization
of the monomer ratios, such as our observation that deprotection of pendent t-butyl esters
only occurred when [PAG] > [glycidyl methacrylate] (GMA), suggesting that epoxide
moieties will to some degree scavenge photo-generated acid.
○ Details along these lines are provided in the new supplementary text section titled
*Rationale for monomer ratios on copolymers 1 and 2*.
- • Are the polymers random (at least looking at the experimental section that seems to be the case)?
Please provide information on polymerization kinetics. Will the nature of the polymer (i.e., random
vs. block vs statistical) impact the fibrillar self-assembly upon UV exposure?
○ Our copolymers are random and were prepared by conventional free radical
polymerization. Incorporation ratios of each monomer were in excellent agreement with the
feed ratios employed.

- ○ We agree with reviewer 2 that block copolymers offer another potentially interesting
platform for mesoscale structures of this type and have a separate line of work that is
examining such structures in different contexts.
- ○ In the main manuscript, we inserted “random” on page 4 (where the polymers are first
introduced) and in the Fig 2a caption.
- • Multiple copolymers with different photoacid generators are reported. The manuscript, however,
does not discuss the fibrillar assemblies of 1 neither does it provide a comparison between the
copolymers. The rationale for using two photoacid generators is unclear. Please discuss the
significance and assembly behavior of copolymer 1.
- ○ We first discovered the photoacross platform with copolymer 1 by patterning with DUV
(~254 nm) and subsequently developed a second pathway with PAG 2 that enabled
photoacrossing at ~365 nm. This has been clarified on page 5 with the phrase that the
photoacross platform was demonstrated with both PAGs 1 and 2, demonstrating the
versatility of the approach across different wavelengths.
- • Fig 4 caption, please move the (a) at the beginning of the caption.
- ○ The caption has been reorganized to improve clarity.
- • Please include page numbers for citation 63.
- ○ This has been corrected and is now reference 64.
- • The ratio of monomers for the three copolymers in synthetic procedure are different from the table
in Fig S1.
- ○ The feed ratios listed in our synthetic procedure are mole equivalents of *feed* ratio
compared to thermal initiator. The numbers a, b, c, and d (now in Figure 2a) give the
*incorporation* ratio of each monomer x100 to give a mole percent of each.
- • The authors investigated the effect of thickness h_0 on the dimension of the formed helices. How
about the width of the alternating domains? Is there a strict requirement such as a specific value of
a ratio for the width of the irradiated domain and masked domain? As far as I am concerned, this
would play a bigger role than thickness on determining the helical pitch and radius.
- ○ We suspect that the reviewer is pointing to the relative *length* of the rigid (masked)
segments (l_s) compared to the length of the photoacross (l_c). We have modified Figure 3a
and included additional text on page 7 to better highlight these dimensions. The general
point that spacing between photoacrosses, their angle ϕ and their fold-angle per acrosses is
consistent with our geometric model for programming the resulting helical geometry ,
shown schematically in Fig. 3a. As a test of this basic control, we have focused on here
the role of ϕ and thickness in determining helix dimensions, but we fully agree with the
reviewer that alteration of lateral spacing and width of the photoacrosses will also present
additional means of shape programming.
- ○ Alternatively, the reviewer may be inquiring about the width dimension (w) as compared to
thickness (h), as depicted in Figure 1 (left). In this case, we can consider the relative role
of w and h in determining photoacross bending by examining the bending mechanics of an
elastic beam. Since bending stiffness scales with h^3 and w^1 , h plays a much larger role
than w in determining extent of bending. Moreover, we expect the bending moment applied
by gradient swelling through the photoacross to scale linearly with w . Accordingly, given
linear dependence of both stiffness (which resists bending) and bending moment (which
drives bending), the width-dependent terms cancel so that photoacross bend angle (and
resulting helical pitch and radius) are independent of width.
- • The helicity of Fig 5f is right-handed while the helices in Fig 5a-e are left-handed. Is this a mistake?
- ○ We thank the reviewer for catching this and we have updated the schematics of Fig. 5f to
reflect left-handed chirality. We note that the dimensional limits highlighted by these
schematics are independent of handedness, and we have updated the Figure 5 caption to
reflect this.
- • Why is the dispersity for copolymer 1 so much broader than 2 and 3?
- ○ The thermal initiator was used at a higher initial concentration in copolymer 1 than
copolymers 2 and 3 (see synthetic protocols). We expect that this resulted in a higher
concentration of active propagating radical chain ends over the course of the
polymerization, leading to a higher rate of radical termination and a broader dispersity.

Reviewer #3 (Remarks to the Author):

The manuscript reports on, allegedly, the first synthetic mesoscale fiber/ply/rope assembly. This is a
synthetic equivalent to the numerous natural examples of fiber assemblies (collagen, keratin, and many
others).

A lot is said in the manuscript on how to achieve these fiber bundles/ropes and the geometrical properties
of the obtained structures. Few is told about their mechanical properties. Being a mechanical engineer, I
am much more interested in the later (mech. prop.) than in the chemical methods used to obtain these
ropes.

- • We appreciate this comment as it highlights the advances underpinning this work. While
topologically linked filamentous structures are well-understood to be critical for the advantageous
mechanical properties of textiles and ropes (references 27-30, 63), our primary goal is not to
advance the properties of these widely-known and widely-used material architectures. Rather, the
engineering challenge that we address is *how to create these remarkable structures* through a
bottom-up, autonomous process. To the best of our knowledge, our manuscript is the first to
develop a generalizable method for the coordinated actuation of programmable filaments that direct
the self-spinning assembly of textile-like microfilament plies. We achieve this by programming
segmental mechanics within *mesoscale filaments*. We go beyond previously established
mechanical processes by building in spatio-temporal control that is necessary for creating
topologically linked structures.
- • While we maintain that the *fabrication pathway* to self-spun mesoscale fibers is the major
intellectual contribution of this manuscript, and generalizable beyond the specific chemistry we
have chosen, we have considered the reviewer's comments seriously and have included
experiments that highlight the differences in elastic behavior between single-ribbon helices versus
multi-ribbon bundles, and different values of ϕ .
- • To improve the manuscript by clarifying our primary contributions and advances, as well as adding
demonstrations of the mechanical response of the assembled structures, we have:
 - ○ Updated the text in our introductory section (page 3) to emphasize that the main innovation
of this work is a *novel route to self-assembled mesoscale filaments underpinned by*
*generalizable principles*
 - ○ Added reference 49, a review that describes known strategies, including in-plane
differential swelling, to program 3D shape morphing
 - ○ Added new sentences on page 7 (and a reference to Fig. S1) to better illustrate the
photocrease bending mechanism and the broader generalizability of our "local control of
curvature and twist" strategy.
 - ○ Added text to the conclusion on page 17.
 - ○ Referenced supporting videos S9 and S10 (included with the originally submitted
manuscript)
 - ○ Added new supporting videos S11 and S12
 - ○ Added new Figure (S20) that describes the experiment (Fig. S20a-c) and the mechanical
impact of both ϕ and n (Fig. S20d)
 - ○ Added new supplementary text section entitled *Comparing helix and bundle mechanics by*
*bending in flow* that outlines the experiment and conclusions

For the reader from outside Polymer Chemistry, the main query that arises on the fabrication scheme is
how symmetry breaking is achieved: how the photocreasing and the dynamics of the swelling eventually
produce 'bottom-face-in' bending and not the other way around.

- • We thank the reviewer for bringing up this point for clarification. Symmetry breaking is achieved via
a swelling gradient through the photocrease thickness, with greater swelling near the top surface,
following along well established mechanisms of programmable buckling of soft material films (as
reviewed in ref. 49). This is attributed in the original manuscript to attenuated UV intensity through
the MSP thickness, giving rise to a gradient in released photoacid (see the top of page 7 and the
final paragraph of page 11). To emphasize and clarify this, we made the following changes:

- ○ Created a new supporting Figure (S1) to describe this through-thickness swelling gradient.
- ○ Added new sentences on page 7 to enhance the description of attenuated UV radiation
- through the photocrease thickness, which affords a gradient in deprotection of pendent *t*-
- BOC groups to control the gradient swelling process.
- ○ Added text and a reference to Figure S1 in the “mechanistic insights” section on page 10
- and 11.

I think I understood it, but would not bet my salary that I got it right! After all, one wants to understand what
is the mechanical force that drives the helix formation. The reader is left with the question: are these
synthetic ropes mechanically better than natural fibril assemblies, or are they chemically more stable, less
prone to aging or acid attacks, or...

- • The reviewer’s point is well-taken and yet we emphasize that the focus of our manuscript is on the
self-bundling mechanisms and the control over variably inter-linked structures in the resulting multi-
filament plies. We also emphasize that our photocrease platform is based on generalizable
principles, and the advance of self-spinning mesoscale filaments need not be confined to specific
polymer chemistries that we have demonstrated here. We are interested in further probing and
expanding their properties (e.g. chemical, mechanical, field responsiveness) and yet it would be
premature to go down that path and draw robust conclusions at this stage. Nonetheless, the
question is good and at a minimum we can say that, in accord with the robust polymer structures
used to prepare the ribbons and bundled structures, the photo-generated acid produced in our
process leads to no apparent degradation of the individual ribbons or the bundled structures.
Moreover, the bundled structures are maintained in fluids for long periods of time (at least months
but likely much longer) and thus there is excellent potential for achieving structures for long-term
use. While our objective is not to make materials that are necessarily mechanically superior to
those found in nature, the opportunity lies ahead to probe such properties.

remarks:

(a) authors might want to insist on the fact that all this is done in an aqueous solution. The average reader
outside the field of Polymer Chemistry might not get this at first.

- • The updated text on page 7 now emphasizes that the swelling-mediated photocrease actuation is
due to the introduction of aqueous solution.
- • The updated text on page 10 now also connects photocrease bending to the introduction of
aqueous solution.

(b) Figure 3b. I get the feeling that a better choice for the axes would make the plot more
interesting/universal. Why not plot pitch times tan phi against radius + thickness for example?

- • To maximize the information provided to the reader, we have added the suggested plot in the
Supplementary Information as Figure S4 and have added a reference to this plot in the main text
on page 9. While we agree that this change to the y axis elegantly collapses the data (of course,
consistent with the governing of helices and our assumption of constant photocrease folding), our
intent with Figure 3b is to communicate that we can independently tune pitch and radius, over a
wide and ultimately consequential range for bundling, by using ϕ and h .

REVIEWERS' COMMENTS

Reviewer #2:

The manuscript has improved significantly and I appreciate the careful repossess to my comments. I am happy to recommend acceptance of the manuscript. Two very minor comments:

It would be helpful if Figures S14 and S16 also included the 'blow up region' shown in Figure S15.

I am always reluctant to sue 'novel' or 'new' in an article. If it is not novel or new why publishing it. I recommend removing these words from the text.

Reviewer #3 (Remarks to the Author):

The concerns I had on the first version of the manuscript have been addressed in the revision.

The manuscript is now clearer and show interesting results on the engineering of microscopic fiber bundles.

**REVIEWER COMMENTS**

Reviewer #2: The manuscript has improved significantly and I appreciate the careful repossess to my
comments. I am happy to recommend acceptance of the manuscript.

**We thank reviewer 2 for this last note and are delighted at their recommendation**

Two very minor comments: It would be helpful if Figures S14 and S16 also included the 'blow up region'
shown in Figure S15.

**Supplementary Figures 14 and 16 have been modified with insets that zoom in on peak-dense regions of**
**the spectrum, consistent with Supplementary Figure 15.**

I am always reluctant to sue 'novel' or 'new' in an article. If it is not novel or new why publishing it. I
recommend removing these words from the text.

**Per the reviewer's recommendation, we have removed all instances of the words "novel" and "new."**

Reviewer #3 (Remarks to the Author):

The concerns I had on the first version of the manuscript have been addressed in the revision.
The manuscript is now clearer and show interesting results on the engineering of microscopic fiber bundles.

**We thank reviewer 3 for their thoughtful commentary and guidance; we are delighted that their concerns**
**have been addressed in this draft.**